# From network to phenotype: the dynamic wiring of an Arabidopsis transcriptional network induced by osmotic stress

Lisa Van den Broeck[1,2,†] iD, Marieke Dubois[1,2,†,‡] iD, Mattias Vermeersch[1,2], Veronique Storme[1,2], Minami Matsui[3] & Dirk Inzé[1,2,*] iD

## Abstract

Plants have established different mechanisms to cope with environmental fluctuations and accordingly fine-tune their growth and development through the regulation of complex molecular networks. It is largely unknown how the network architectures change and what the key regulators in stress responses and plant growth are. Here, we investigated a complex, highly interconnected network of 20 Arabidopsis transcription factors (TFs) at the basis of leaf growth inhibition upon mild osmotic stress. We tracked the dynamic behavior of the stress-responsive TFs over time, showing the rapid induction following stress treatment, specifically in growing leaves. The connections between the TFs were uncovered using inducible overexpression lines and were validated with transient expression assays. This study resulted in the identification of a core network, composed of ERF6, ERF8, ERF9, ERF59, and ERF98, which is responsible for most transcriptional connections. The analyses highlight the biological function of this core network in environmental adaptation and its redundancy. Finally, a phenotypic analysis of loss-of-function and gain-of-function lines of the transcription factors established multiple connections between the stress-responsive network and leaf growth.

**Keywords** growth regulation; mild osmotic stress; short-term stress response; transcription factors; transcriptional network

**Subject Categories** Genome-Scale & Integrative Biology; Plant Biology; Transcription

**Mol Syst Biol. (2017) 13: 961**

## Introduction

Plant growth is a very complex quantitative trait and depends on both the genetic background and environmental conditions that can stimulate or adversely affect growth (Doust *et al*, 2014; Saïdou *et al*, 2014). Each environmental stimulus causes a specific response established by multiple regulatory components forming an interconnected network rather than a linear pathway (Vermeirssen *et al*, 2014; Miao *et al*, 2015; Luo *et al*, 2016). In addition, environmental changes are often multifactorial, such as heat and drought often occurring simultaneously. The combination of different environmental signals thus leads to complex responses, which are integrated by gene regulatory networks (GRNs) that enable the regulation of complex traits such as growth. It is therefore necessary to study these genetic networks as one entity in addition to studying the role of their individual components in order to get insights into the arising phenotype.

A GRN can be defined as a combination of regulatory proteins such as transcription factors (TFs) that function together to regulate a specific set of output genes. A very well-known example of a GRN is the circadian clock regulatory network (Nagel & Kay, 2012; Pokhilko *et al*, 2012; Seaton *et al*, 2015; Hernando *et al*, 2017). This network consists of a core oscillator module of three TFs (CIRCADIAN CLOCK ASSOCIATED1 (CCA1), LATE HYPOCOTYL (LHY) and TIMING OF CAB1 (TOC1)) that forms the base of a larger interconnected network regulating circadian rhythms, hypocotyl growth, and flowering of Arabidopsis plants through transcriptional but also post-translational regulation, chromatin remodeling, and alternative splicing (Nakamichi, 2011; Malapeira *et al*, 2012; Perez-Santángelo *et al*, 2013; Wang & Ma, 2013). The core circadian clock network in Arabidopsis has even been extrapolated to crops such as rice, maize, soybean, and *Brassica rapa* (Murakami *et al*, 2007; Liu *et al*, 2009; Xu *et al*, 2010; Wang *et al*, 2011). A more specific example of a smaller GRN is the BRASSINAZOLE RESISTANT(BZR)—PHYTOCHROME INTERACTING FACTOR 4 (PIF4)—DELLA module that

1 Department of Plant Biotechnology and Bioinformatics, Ghent University, Ghent, Belgium
2 VIB Center for Plant Systems Biology, Ghent, Belgium
3 RIKEN Center for Sustainable Resource Science, Kanagawa, Japan
  *Corresponding author. Tel: +32 9 331 38 06; E-mail: dirk.inze@psb.vib-ugent.be
  †These authors contributed equally to this work
  ‡Present address: Institut de Biologie Moléculaire des Plantes, CNRS, Strasbourg, France

integrates brassinosteroid, light, and gibberellin signals to regulate cell elongation (Bai *et al*, 2012; Claeys *et al*, 2014a; Zhiponova *et al*, 2014). Environmental signals disturb the molecular steady state of GRNs by changing the gene expression levels or by post-translational modifications triggering the (de)activation of a protein. Under such changing conditions, networks dynamically evolve to reach a new steady state in which the components are in balance. At the phenotypic level, the modifications in the GRN ultimately lead to a particular output, for example, growth stimulation or inhibition. The existence of such complex networks facilitates the fine-tuning of the response to a continuously varying input, such as heat or drought stress.

The compound mannitol is used in plant research as a molecule to induce osmotic stress and interfere with plant growth (Claeys *et al*, 2014b). Low concentrations of mannitol (25 mM) induce mild stress, triggering a decrease in Arabidopsis rosette size of approximately 50% without affecting the development or survival. Therefore, this setup can be used to investigate the molecular mechanisms underlying leaf growth inhibition (Skirycz *et al*, 2011; Claeys *et al*, 2014b). During Arabidopsis leaf development, the growth of an emerging leaf primordium is first solely driven by cell proliferation, resulting in an increased cell number. After a few days, cells at the distal end of the leaf exit the mitotic cell cycle and start to expand and subsequently differentiate (Donnelly *et al*, 1999; Andriankaja *et al*, 2012). At this point, growth is merely driven by cell expansion and, in the epidermis, by the division activity of meristemoid cells (White, 2006; Andriankaja *et al*, 2012; Gonzalez *et al*, 2015). Both cell proliferation and cell expansion can be adversely affected by mild osmotic stress conditions (Skirycz *et al*, 2011; Huber *et al*, 2014). Mannitol-induced stress inhibits the cell cycle by a two-step process called the "pause-and-stop" mechanism (Skirycz *et al*, 2011). In the first phase, the "pause" phase, the cells are kept in a latent state allowing rapid resumption of the cell cycle when conditions are again favorable. When the osmotic stress persists, the cells permanently exit the cell cycle and differentiate, called the "stop" phase.

Previously, a transcriptome analysis on microdissected, actively growing leaf tissue exposed to low concentrations of mannitol was performed to identify putative molecular players orchestrating the observed growth arrest (Skirycz *et al*, 2011). Upon short-term exposure to mannitol, a gradually increasing number of genes encoding TFs is significantly upregulated, suggesting that a transcriptional cascade initiates the early response to mannitol. Few members of this transcriptional cascade have been studied previously, such as the rapidly induced *ETHYLENE RESPONSE FACTOR 6* (*ERF6*), which activates the expression of *GIBBERELLIN2-OXIDASE6* (*GA2-OX6*), a gene encoding a gibberellin-inactivating enzyme (Rieu *et al*, 2008; Dubois *et al*, 2013). Because of the resulting lower levels of gibberellin, DELLA proteins are stabilized, which ensures that cells permanently exit the cell division phase and are pushed to cell differentiation (Claeys *et al*, 2012). The transcriptional repressor ERF11 also has been characterized and could counteract the effect of ERF6 both on molecular and phenotypic level (Dubois *et al*, 2015).

In this study, we investigated a subset of mannitol-responsive TFs and show that they form a dense GRN that is very rapidly induced upon mannitol treatment. We demonstrate the transcriptional connections between these individual components and give new insights into their regulatory capacities on the expression of target genes. Using this systems biology approach, we identified a hub of five TFs (ERF6, ERF8, ERF9, ERF59, and ERF98) that drives most regulatory connections. Finally, we studied the role of the 20 TFs in the regulation of leaf growth under standard conditions and when exposed to mild osmotic stress, leading to the identification of multiple growth-regulating TFs.

# Results

## A GRN of 20 TFs is specifically activated in growing leaves exposed to mannitol

A previous transcriptome analysis upon short-term exposure of Arabidopsis seedlings to mannitol has identified genes that are rapidly induced upon osmotic stress in young proliferating leaves (Skirycz *et al*, 2011). Among them, ERF6 appeared to play a key role in this early stress response, enabling the inhibition of leaf growth and the simultaneous activation of stress-inducible genes. Based on the identified mannitol-responsive genes (Skirycz *et al*, 2011) and the ERF6 target genes (Dubois *et al*, 2013), we selected 28 genes encoding TFs with a putative role in the mannitol-mediated growth retardation. To measure the transcriptional induction of these 28 genes by mannitol, 15-day-old plants grown on half-strength Murashige and Skoog (1/2 MS) medium covered with a nylon mesh were transferred to medium containing 25 mM mannitol or control medium (Skirycz *et al*, 2011). After 4 h, the third leaf was harvested for transcript profiling. At this stage, the third leaf is actively growing and mostly contains expanding cells. Because the transcriptional induction was confirmed for 20 genes (Appendix Fig S1), we hypothesized that these 20 TFs could act together in a transcriptional network to regulate growth upon stress.

Half of the TFs of the putative mannitol-responsive GRN belong to the ERF family (Appendix Table S1) (Nakano *et al*, 2006; Skirycz *et al*, 2011; Phukan *et al*, 2017), containing a single AP2/ERF domain that is responsible for the specific binding to GCC boxes in the promoter of their target genes (Fujimoto *et al*, 2000; Yang *et al*, 2009). Three ERF proteins, ERF8, ERF9, and ERF11, belong to group VIII and are putative transcriptional repressors, because they contain an ERF-associated amphiphilic repression (EAR) domain (Nakano *et al*, 2006). Six other stress-induced ERFs belong to group IX: ERF-1, ERF2, ERF5, ERF6, ERF59, and ERF98. ERF5 and ERF6 contain an additional motif, CMIX-5, which is a predicted phosphorylation site (Fujimoto *et al*, 2000; Nakano *et al*, 2006). The last ERF protein part of the putative mannitol-induced network, RAP2.6L, belongs to group X (Nakano *et al*, 2006). Seven members of the proposed GRN are part of the WRKY TF family: WRKY6, WRKY15, WRKY28, WRKY30, WRKY33, WRKY40, and WRKY48, which contain a conserved sequence (WRKYGQK) followed by a zinc finger motif, enabling the binding to DNA at the position of a W-box TTGAC(C/T) (Wu *et al*, 2005). Finally, three other TFs, ZAT6 and STZ, belonging to the Zinc Finger TF family (Englbrecht *et al*, 2004; Ciftci-Yilmaz & Mittler, 2008; Kiełbowicz-Matuk, 2012), and MYB51 (Stracke *et al*, 2001; Dubos *et al*, 2010), are part of the proposed mannitol-inducible network (Appendix Table S1).

To investigate the developmental timing of the putative GRN into more detail, we measured the expression level of the 20 genes upon

stress in the third leaf of wild-type plants during the proliferating (9 days after stratification [DAS]), expanding (15 DAS) and mature (22 DAS) developmental stage (Dataset EV1). With the exception of *ERF8* and *ERF9*, which were most probably only transiently induced by mannitol, all other 18 TFs were significantly upregulated under stress conditions (Student's *t*-test, FDR < 0.05) in proliferating or expanding tissue (Fig 1). For about half of these genes, the level of induction in proliferating and expanding tissue was similar. Three genes, *ERF5*, *ERF6*, and *ERF11*, were induced more highly in expanding leaf tissue, whereas six genes, *ERF59*, *ERF98*, *MYB51*, *WRKY6*, *WRKY30,* and *WRKY40*, were induced more strongly in proliferating leaf tissue. Interestingly, none of the TFs were significantly upregulated in mature leaf tissue (Fig 1), suggesting that the putative stress-responsive GRN is only induced in growing leaves, because these tissues are prone to growth inhibition upon mild stress.

## The GRN shows the sequential activation of four TF groups

Because expression analysis has previously shown the early upregulation of these genes upon mannitol treatment (Skirycz *et al*, 2011), the young developing third leaf (15 DAS) was harvested at a high temporal resolution (20 min, 40 min, 1 h, 2 h, 4 h, 8 h, 12 h, 16 h, 24 h, and 48 h) after transfer to control or 25 mM mannitol-containing medium. RNA was extracted, and a detailed expression pattern over time for each gene of the putative GRN was generated with the nCounter Nanostring® technology (Dataset EV1). This technology enables the determination of the expression level of multiple genes in parallel without losing sensitivity.

Within 1 h upon stress, nine of the 20 TF-encoding genes were significantly upregulated (Table EV1; Student's *t*-test, FDR < 0.05) and most genes reached a maximum expression level after 2 h, demonstrating the very rapid response of this regulatory network. When considering the earliest time points in more detail, the initial upregulation was not equally fast but instead occurred in a sequential manner (Fig 2). The TFs could be classified into four different groups based on the initial time point at which their expression exceeded the threshold of $\log_2$(fold change [FC]) > 1 (Fig 2). The first group included seven genes (*ERF5*, *ERF6*, *ERF11*, *ERF98*, *WRKY40*, *STZ* and *ZAT6*). All genes showed a fast and strong induction, exceeding the threshold already at 40 min (Fig 2A). The second group, including *ERF-1*, *ERF2*, *WRKY30*, *WRKY33,* and *MYB51*, was upregulated from 1 h onward (Fig 2B). However, the induction of these genes, except for *WRKY30*, was not as strong as that of the first group; the genes of the second group reached a maximum of approximately $\log_2$(FC) 4 compared to a maximum of approximately $\log_2$(FC) 6 in the first group. The third group, which passed the threshold at 2 h, contained *WRKY6*, *WRKY15*, *WRKY28*, *WRKY48*, *ERF59,* and notably two genes encoding the repressors ERF8 and ERF9 (Fig 2C). The induction was even less strong than that of the second group; most genes reached a maximum around $\log_2$(FC) 3. In the fourth group, the expression of the activator *RAP2.6L* was upregulated only 4 h after mannitol treatment with a maximum of approximately $\log_2$(FC) 5 (Fig 2D).

During later time points (12 h, 16 h, 24 h, and 48 h), three scenarios could be observed (Appendix Fig S2). Following the initial induction, the expression of the TF either (i) gradually decreased to the expression level in control conditions and was not significantly

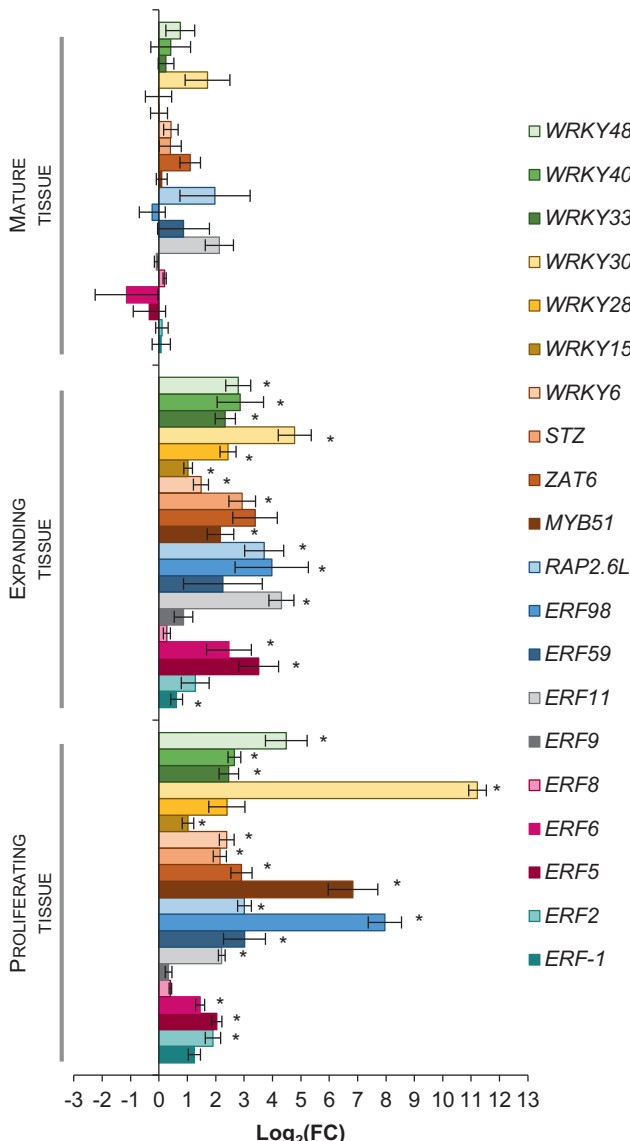

**Figure 1. Mannitol-induced transcriptional changes of the selected TFs in proliferating, expanding, and mature leaf tissue.**

The expression of the 20 genes encoding TFs was measured 24 h after mannitol treatment during the proliferating (*n* = 192 plants), expanding (*n* = 16 plants), and mature (*n* = 16) leaf developmental stage. Expression levels in wild-type plants transferred to mannitol-induced stress were compared to those transferred to control conditions at the same developmental stage.

Data information: Data are presented as mean ± SEM, *n* = 4 independent experiments. FC = fold change. *FDR < 0.05, unpaired two-sided Student's *t*-test.

upregulated at 48 h (Appendix Fig S2A), (ii) reached a minimum and increased again (Appendix Fig S2B), or (iii) remained induced until at least 48 h after stress (Appendix Fig S2C). In total, 11 TFs were significantly upregulated upon 48 h of stress.

In conclusion, the 20 selected TFs were rapidly upregulated upon mannitol treatment and, interestingly, their induction could be divided into four groups of initial transcriptional activation. For most TFs, the maximum expression level was reached after 2 h. Remarkably, the expression of 11 genes remained higher even after

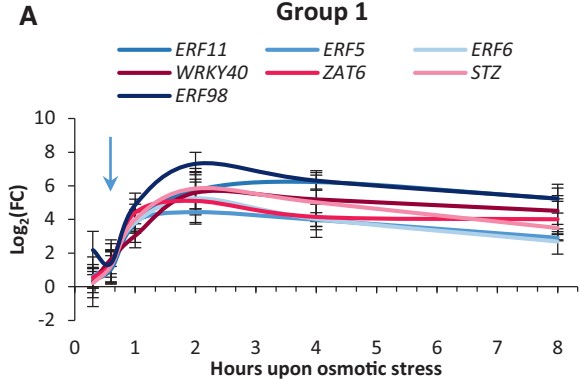

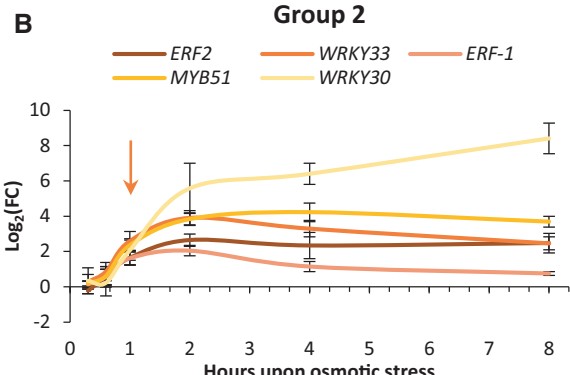

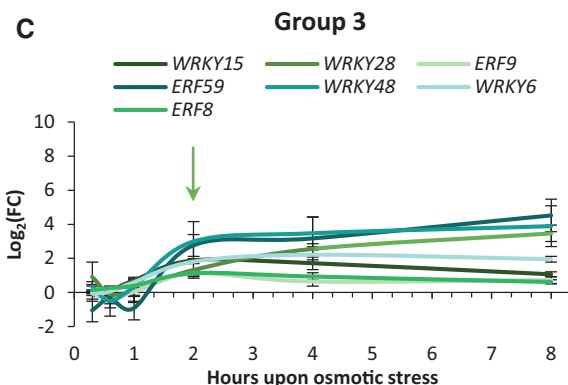

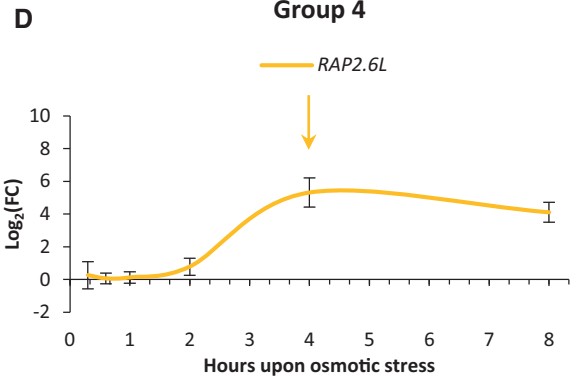

**Figure 2. Four groups of transcriptional induction upon exposure to mannitol.**

A–D Based on a threshold of $\log_2(FC) > 1$, the 20 TFs were categorized into four groups. The first group contains TFs that reached the $\log_2(FC)$ threshold 40 min after mannitol treatment (A), the second group reached the threshold after 1 h (B), the third group after 2 h (C), and the fourth group after 4 h (D). The arrow indicates the initial upregulation of every group.

Data information: Data are presented as mean ± SEM. $n = 4$ independent experiments. FC = fold change. FDR values are available in Table EV1.

48 h of mannitol treatment, suggesting that these TFs also play a role in the long-term response to osmotic stress.

**The GRN is highly interconnected and dynamic**

To validate our hypothesis that the 20 selected TFs act as a network rather than independently, we aimed to identify and visualize the putative GRN. The putative GRN consists of 20 nodes, representing the 20 TFs, and directed edges between the nodes, indicating the transcriptional regulatory connections. To determine these regulatory connections and thus the edges, we performed a large-scale expression analysis with gain-of-function (GOF) lines. We opted for inducible constructs in which a C-terminal fusion protein of the TF of interest and a glucocorticoid receptor (GR) domain is driven by a constitutive 35S promoter. Such fusion proteins reside in the cytosol and can only translocate to the nucleus in the presence of dexamethasone (DEX), enabling the TF to regulate its downstream target genes (Corrado & Karali, 2009). Per TF, two or three independent GOF lines with intermediate or high overexpression of the TF were obtained (Appendix Figs S3–S22), with the exception of three genes (*WRKY6*, *WRKY30,* and *WRKY40*) for which we could not obtain a proper overexpression line. To get an indication of which genes are direct or indirect targets of the induced TF, we opted for a time-course approach rather than an inhibition of translation by cyclo-heximide, because the latter already induced 18 of the 20 TFs by itself (Appendix Fig S23, Hruz *et al*, 2008). Therefore, one independent GOF line was selected for all 17 TFs and transferred at 15 DAS to DEX-containing medium and the third leaf was harvested at 1 h, 2 h, 4 h, 8 h, and 24 h after transfer (Appendix Table S2). The expression of each of the 19 other TFs was measured with nCounter Nanostring (Source Data for Fig 3) (Geiss *et al*, 2008). The time-course experiment gives an indication of whether a gene is putatively a direct, and thus induced during the early time points, or an indirect target of the induced TF.

The expression analysis rendered, for each time point, a network of which the edges are based on the differentially expressed genes in every GOF line (Fig 3A). For example, a directed edge from ERF6 to STZ means that *STZ* was significantly differentially expressed (FDR < 0.1) in the ERF6-GR line at that specific time point and could thus be directly or indirectly regulated by ERF6; *STZ* is then defined as a target gene of ERF6. When considering all time points, we could observe that in nine GOF lines, ERF-1-GR, ERF2-GR, ERF6-GR, ERF8-GR, ERF9-GR, ERF59-GR, ERF98-GR, WRKY15-GR, and WRKY48-GR, the expression of at least half of the other TFs was affected (log[FC] > 1) (Appendix Figs S3–S22). The large amount of observed regulatory interactions clearly demonstrates that the selected TFs form a highly interconnected GRN.

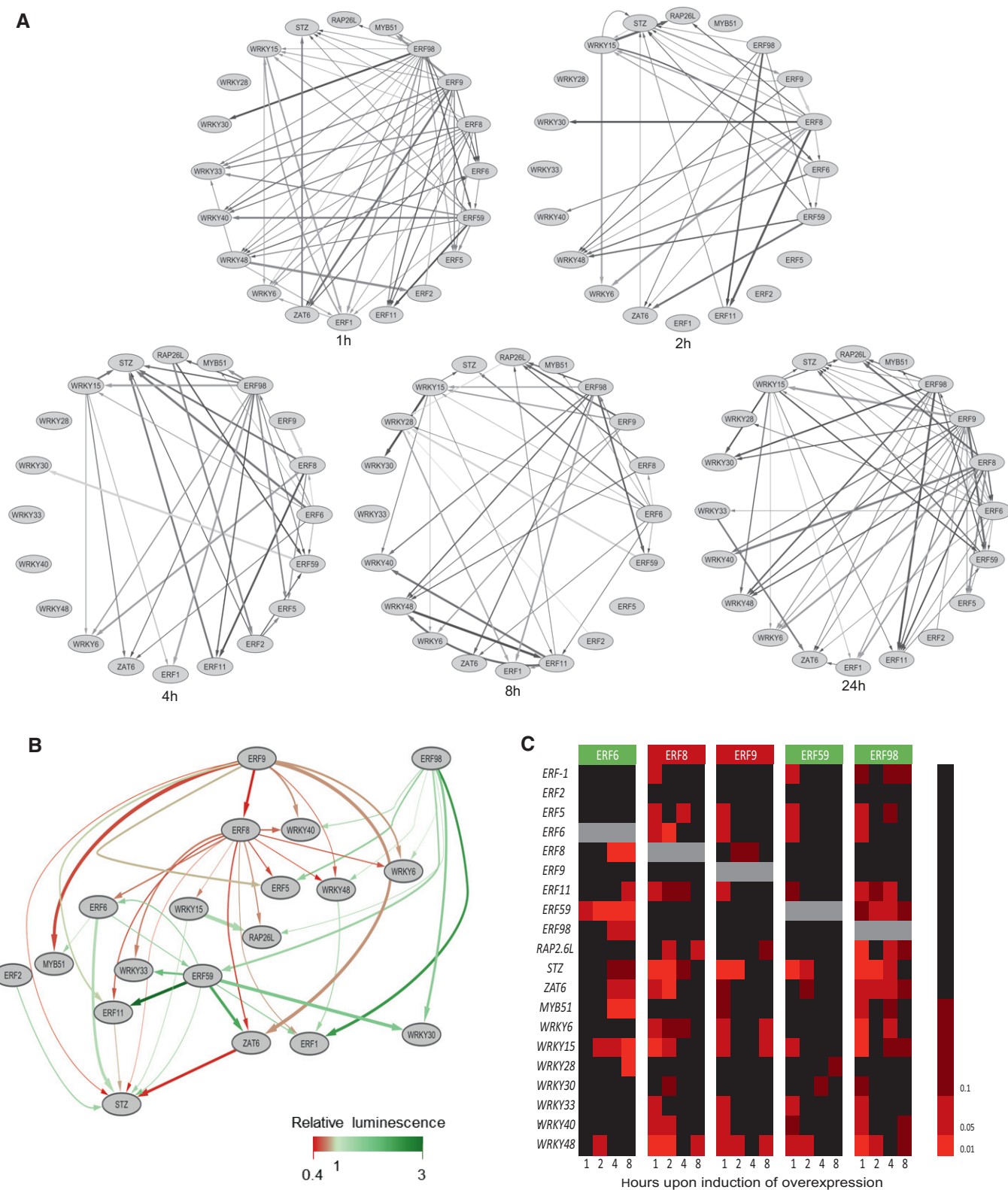

**Figure 3.**

**Figure 3. The regulatory connections of the osmotic stress-responsive GRN.**

A The significant regulatory interactions identified by nCounter Nanostring at 1 h, 2 h, 4 h, 8 h, and 24 h after induction of overexpression of a TF.

B The confirmed regulatory interactions between the 20 TFs part of the GRN, according to transient expression assays (*n* = 4 biological repeats). Green arrows represent activation and red arrows repression.

C Heatmap of the significant regulations upon induction of overexpression of the five members of the core network, the activators (green) ERF6, ERF59, ERF98, and the repressors (red) ERF8 and ERF9. Color code represents FDR-corrected *P*-values with thresholds at FDR = 0.01, 0.05 and 0.1.

Data information: In (A), data are extrapolated from estimated averages, *n* = 3 independent experiments, FDR-corrected *P* < 0.1 (mixed model analysis, user-defined Wald tests). The thickness of the arrows represents the FDR value. In (B), data are presented as averages, *n* = 3 independent experiments. The intensity of the color of the arrows represents the strength of the regulation according to the TEA values and the thickness the FDR value of the nCounter Nanostring experiment. In (C), data are represented as FDR-corrected *P*-values, *n* = 3 independent experiments (mixed model analysis, user-defined Wald tests).

Source data are available online for this figure.

The inclusion of multiple time points allowed to explore dynamic changes in regulatory connections. If we assume that every TF acts directly on its target genes without being influenced by other TFs, we could expect that the continuous induction of overexpression leads to a fast induction followed by a sigmoidal expression pattern of the target genes. For example, the strong activation of *WRKY15* led to the gradually increased expression of part of its target genes such as *STZ*, *WRKY6*, *WRKY30*, *WRKY40*, *ERF-1,* and *ERF11* (Fig EV1A). However, some genes showed an oscillating pattern upon WRKY15-induced activation, such as the target genes *ERF6*, *RAP2.6L*, and *ZAT6* (Fig EV1B). The oscillation of some transcripts was also visible at the network level: most interactions were formed after 1 h of induction and decreased after 2 h or 4 h, but increased again after 8 h or 24 h (Fig 3A, Source Data for Fig 3). The oscillations further strengthen the hypothesis that multiple TFs regulate the expression of the same target gene, leading to multiple indirect effects. The highly fluctuating regulations also emphasize the need for short-term analysis because the steady state of the network masks these connections.

To analyze the transactivation capacities of the TFs on their target genes, the edges based on the transcriptome data at 1 h, 2 h, and 4 h (in total 81 edges) were further verified with transient expression assays (TEAs). Luciferase reporter genes were used to perform TEAs in tobacco (*Nicotiana tabacum*) Bright Yellow-2 (BY2) protoplasts (Vanden Bossche *et al*, 2013). The protoplasts were co-transformed with 35S::TF and pTF::*fLUC* (firefly luciferase) constructs to evaluate whether a TF can activate or repress a target promoter, here defined as the region upstream of the start codon until the next gene with a maximum of 2 kb. In total, 45 out of the 81 edges were confirmed (Appendix Table S3, Appendix Figs S3–S22) and were used to build a more robust GRN (Fig 3B). Two distinct types of edges are represented in the network: red arrows represent inhibition of the expression of the target gene, whereas green arrows represent activation. All TFs were exclusive activators or repressors. For example, ERF8 and ERF9 appeared to be strong repressors, because for all tested target genes, the co-transformation with ERF8 or ERF9 led to a decreased luminescence signal (Fig 3B, Appendix Figs S3–S22). However, it should be noted that the nature of the regulation was not always consistent between the DEX-inducible *in planta* system and the TEA experiments. The repressing function of the literature-described repressors ERF8 and ERF9 (Ohta *et al*, 2001; Nakano *et al*, 2006) seemed to be abolished by fusion with the GR domain, as observed *in planta* (Appendix Figs S3–S22) and in TEAs performed with ERF8-GR or ERF9-GR (Appendix Fig S24). The discrepancy is thus most likely due to the presence of the GR domain close to the EAR motif. The TEAs performed with the

TFs without GR domain are thus more likely to represent the activity of the endogenous TF.

Among the 45 confirmed regulatory connections, 39 were originating from only five ERF genes, *ERF6*, *ERF8*, *ERF9*, *ERF59,* and *ERF98*. We further refer to these TFs as the core network (Fig 3C). In conclusion, the large-scale expression analysis revealed a dense GRN with generally a strong induction of the network genes when one member is activated. More than half of the regulatory connections could be confirmed by an independent transactivation assay and led to the identification of a core network.

**Most TFs are involved in leaf growth regulation**

Because the 20 selected TFs were specifically induced by mannitol treatment in growing leaf tissue, we further characterized their role in leaf growth. For every TF, loss-of-function (LOF) lines were obtained from different collections (SALK, GABI-KAT, FLAG, and SAIL) containing a T-DNA insertion in or near the coding sequence of the gene of interest. This caused abortion or decreased expression of the TF, with the exception of *wrky30*, in which the T-DNA insertion resulted in an increased expression (Appendix Figs S3–S22). Because ERF5 and ERF6 have been described to be redundant (Dubois *et al*, 2013), the double mutant was used for phenotypic analysis.

LOF and GOF lines (35S::TF-GR as described above) were grown *in vitro* on 1/2 MS medium (with addition of 5 μM DEX for the GOF lines) until 22 DAS. Subsequently, leaf series were made to determine the number of rosette leaves and the individual leaf size (Appendix Figs S3–S22, Table EV2). Interestingly, all core network members showed a growth phenotype when knocked out or overexpressed (Table EV2, Fig 4). Among the LOF lines, the knock-out line of the core TF ERF8 showed a significant increase in rosette area of 12% (*P* < 1E-7, Tukey's test) compared to the wild type (Fig 4A). In addition to *erf8*, also *erf11* and *wrky30* showed a significant increase in rosette area of 8% (*P* = 0.012, Tukey's test) and 13% (*P* = 2.2E-5, Tukey's test), respectively. On the contrary, *erf2*, *wrky6*, *wrky15*, *rap2.6L,* and *stz* had smaller rosette areas. Intriguingly, we observed a difference concerning the position of the most affected leaves in some of the mutants: the rosette size reduction in the *rap2.6L* line was caused by a reduced area of the younger leaves, whereas in *myb51* the older leaves were smaller (Fig 4B). The other core network members had a growth phenotype when being overexpressed. *ERF59* overexpression caused a growth stimulation (increased rosette area) of 26% and 17% in two independent lines (Fig 4C and D). In addition to the already characterized ERF6-GR dwarfed phenotype (Fig 4C;

Dubois *et al*, 2013), the most striking growth phenotype of the GOF lines was observed in all three independent ERF9-GR lines, which showed a size reduction of 30%, 57% and 37% (Fig 4C and D). ERF98-GR ($-12\%$, $P = 6.2E\text{-}6$, $-11\%$, $P = 1.E\text{-}4$, $-38\%$, $P < 1E\text{-}7$, Tukey's test) showed a less drastic but significant decrease in rosette area (Fig 4C and D). Other TFs that showed a decreased rosette size in all tested independent lines were ERF11-GR and WRKY15-GR (Fig 4C and D). Similar to the LOF lines, not all leaves were equally affected in the GOF lines: in both ERF11-GR lines, the growth reduction was more pronounced in the younger leaves (Appendix Fig S9), whereas in two ERF98-GR lines, the growth reduction was more visible in the older leaves (Appendix Fig S11). None of the LOF or GOF lines showed a significant difference in leaf number, indicating that the changes in rosette area are entirely the result of an altered leaf size. In conclusion, 12 of the 20 TFs could affect rosette size when overexpressed or knocked down (Fig 4, Table EV2), confirming the hypothesis that most TFs of the GRN have a growth-regulating function. The maximum increase and decrease in rosette area were observed in two GOF lines of the core network, ERF59-GR (+26%) and ERF6-GR ($-92\%$), respectively. Moreover, we could identify four of the core TFs (ERF6, ERF8, ERF9, and ERF98) as negative growth regulators and one, ERF59, as a positive growth regulator (Table EV2).

The transgenic lines showing a growth phenotype were further subjected to a detailed cellular analysis to evaluate whether the observed growth difference results from an altered cell number or cell area. The increased leaf area of *erf8* and decreased leaf area of *wrky15* and *stz* were the result of a change in cell number, which was increased in *erf8* (66%, $P = 6.7E\text{-}05$, Tukey's test) and decreased in *wrky15* and *stz* ($-17\%$, $P = 6.3E\text{-}3$ and $-14\%$, $P = 0.13$, Tukey's test, respectively; Fig 4E). Additionally, a cell area decrease for *erf8* and increase for *wrky15* and *stz* could be observed, pointing toward a compensation mechanism for the cell number alternations or a developmental shift. A similar compensation mechanism was observed for WRKY15-GR, WRKY28-GR, ERF98-GR, and ERF9-GR: the observed reduced growth caused by a decreased cell number ($-12\%$, $P = 0.024$, $-31\%$, $P = 7.3E\text{-}4$, $-23\%$, $P = 2.3E\text{-}05$ and $-71\%$, $P = 1.4E\text{-}10$, Tukey's test, respectively) was partially compensated by an enhanced cell expansion (Fig 4F). On the contrary, the enlarged leaf areas for the ERF59-GR and ERF2-GR lines resulted from an increased cell area (21%, $P = 0.014$ and 18%, $P = 3.3E\text{-}3$, Tukey's test, respectively; Fig 4F). Together, we could observe that both cell proliferation and cell expansion could be affected, suggesting that both processes are under control of the GRN, which corresponds to the observed expression patterns of the genes in wild-type plants (Fig 1).

## Perturbing the GRN results in the differential expression of genes involved in gibberellic acid metabolism

Previous reports (Claeys *et al*, 2012; Dubois *et al*, 2013) have shown the importance of gibberellic acid (GA) degradation in the response to mannitol treatment. To further explore this finding in our experimental setup, the expression profile of genes encoding two GA degradation (*GIBBERELLIN2-OXIDASE4* and *GIBBERELLIN2-OXIDASE6*) and two GA biosynthesis enzymes (*GIBBERELLIN3-OXIDASE1* and *GIBBER-ELLIN20-OXIDASE1*) was measured in expanding leaves over time (20 min, 40 min, 1 h, 2 h, 4 h, 8 h, 12 h, 16 h, and 24 h after transfer to mannitol-containing or control medium). We could observe the upregulation of *GA2-OX6*, as previously described in proliferating tissue (Fig 5A; Skirycz *et al*, 2011). Remarkably, in expanding leaf tissue, the upregulation by mannitol reached higher levels ($\log_2(\text{FC}) = 4.71$ compared to $\log_2(\text{FC}) = 0.90$ in proliferating tissue, Dataset EV1) and in addition, the timing of the upregulation was altered. Already 2 h after mannitol treatment, *GA2-OX6* expression levels reached significant values ($\log_2(\text{FC}) = 5.21$, FDR = 0.014, Student's *t*-test) in expanding tissues, whereas the induction was only significant after 24 h in proliferating tissue (Skirycz *et al*, 2011; Claeys *et al*, 2012) (Dataset EV1). In accordance with previous observations (Skirycz *et al*, 2011; Dubois *et al*, 2013), none of the other tested GA oxidases showed a significant upregulation over the analyzed time course. However, after 24 h, a significant downregulation of the rate-limiting GA biosynthesis gene, *GA20-OX1* ($\log_2(\text{FC}) = -1.33$, FDR = 0.014, Student's *t*-test), was observed, pointing toward a possible role for GA20-OX1 in the regulation of the growth reduction induced by mannitol, in addition to the predominant role of GA2-OX6.

We further aimed to understand the putative connection between the mannitol-responsive GA oxidases and the selected TFs. To analyze whether this pathway was perturbed in the GOF lines showing an altered rosette area in at least two independent lines, the expression of *GA2-OX6* and *GA20-OX1* was measured 8 h upon DEX-mediated activation of each TF (Appendix Table S2). As expected, *GA2-OX6* expression was significantly upregulated in the ERF6-GR line ($\log_2(\text{FC}) = 6.12$, FDR = 3.9E-9, mixed model analysis, user-defined Wald tests) (Fig 5B). In the ERF9-GR and ERF98-GR lines, a significant change in expression of both *GA20-OX1* and *GA2-OX6* could be observed, whereas in the WRKY15-GR and ERF11-GR lines, the expression of only *GA2-OX6* was altered, although not significantly in the latter (Fig 5B). The GOF lines with an increased *GA2-OX6* expression (GA degradation) and/or a decreased *GA20-OX1* expression (GA biosynthesis) all showed a reduced rosette area (Fig 4C). To evaluate the transactivation capacities of the TFs on the 2-kbp *GA2-OX6* promoter, TEAs in tobacco protoplasts were performed. Three of the

---

**Figure 4. Phenotypic analysis of loss-of-function (LOF) and gain-of-function (GOF) lines of the TFs under control conditions.** ▶

A  At 22 days after stratification (DAS), leaf series of the LOF lines were made and the rosette area was calculated as the sum of the area of all individual leaves (*n* = 10 plants). The rosette area is presented relative to the corresponding wild type.

B  Average area of the individual leaves of *rap2.6L* and *myb51*, two knockout lines with a smaller average rosette area.

C  Rosette area of two or three independent GOF lines per TF (calculated as the projected area or the sum of the leaves, *n* = 10 plants) germinated and grown on DEX-containing medium. Measurements were performed at 22 DAS relative to the control line. "Independent line 1" is the line with the highest overexpression level.

D  A representative picture of the rosette of GOF lines with significant growth phenotypes in both independent lines at 22 DAS. Scale bar is 1 cm.

E, F  Pavement cell number and area of the third leaf at 22 DAS of LOF lines (E) and GOF lines (F) that showed a significant rosette area phenotype.

Data information: In (A, C), data are presented as mean ± SEM, *n* = 3 independent experiments, \*$P < 0.05$ (Tukey's test). In (B), data are presented as mean ± SEM, *n* = 3 independent experiments, \*$P < 0.05$ (mixed model, partial *F*-tests). In (E, F), data are presented as mean ± SEM, *n* = 3 independent experiments. $^{\cdot}P < 0.1$, \*$P < 0.05$, \*\*$P < 0.01$, \*\*\*$P < 0.001$ (Tukey's test).

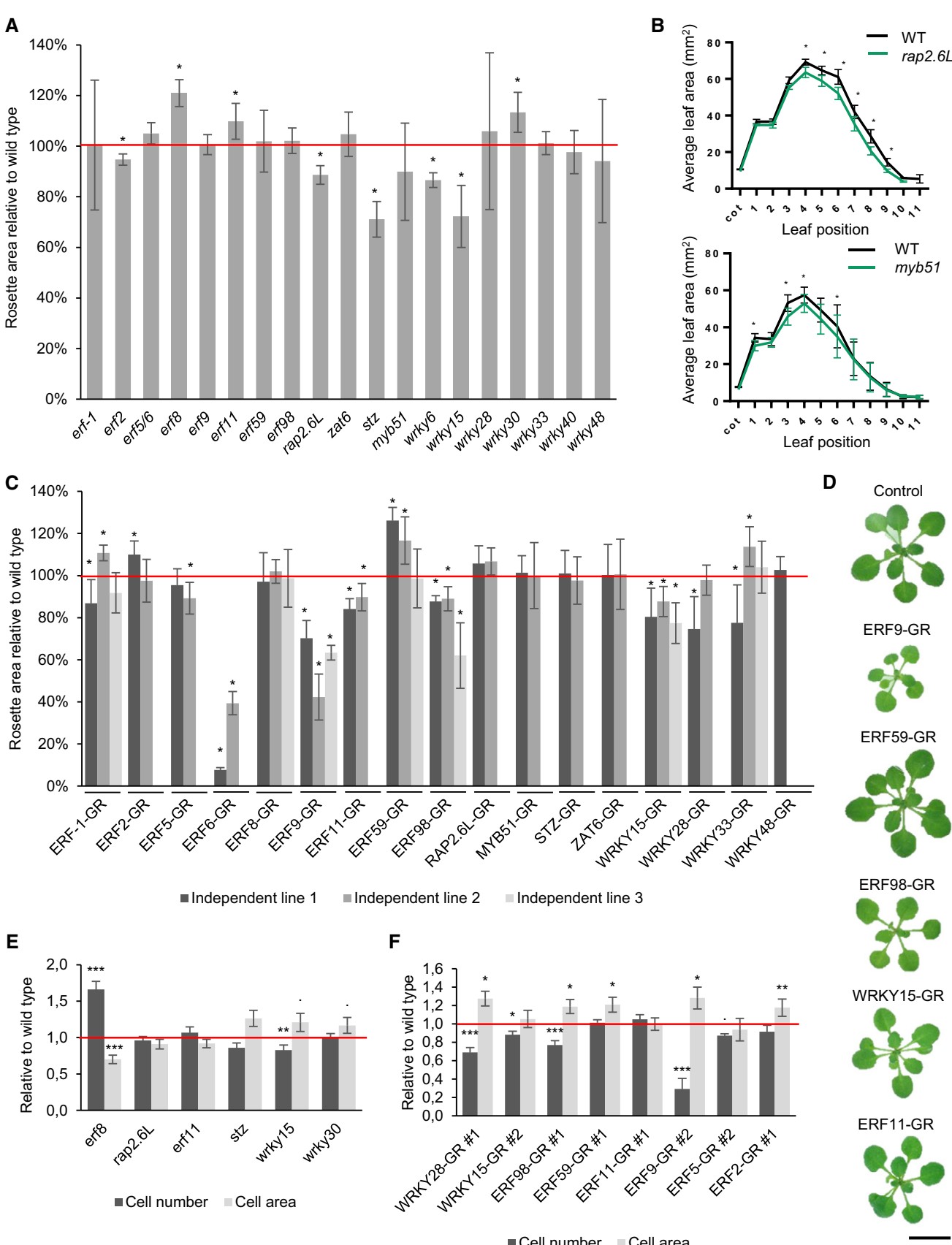

Figure 4.

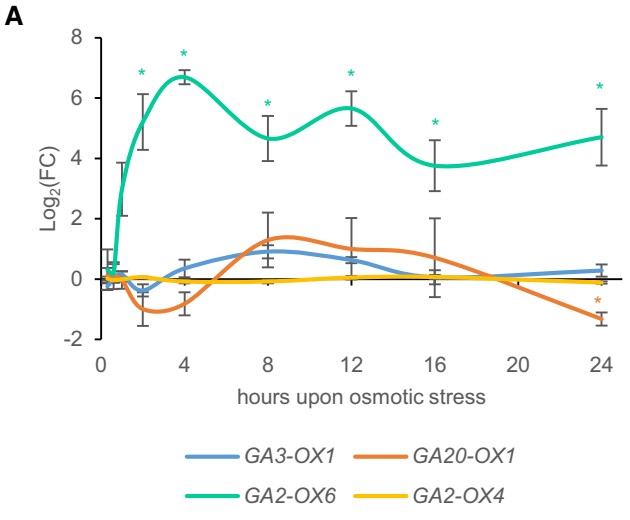

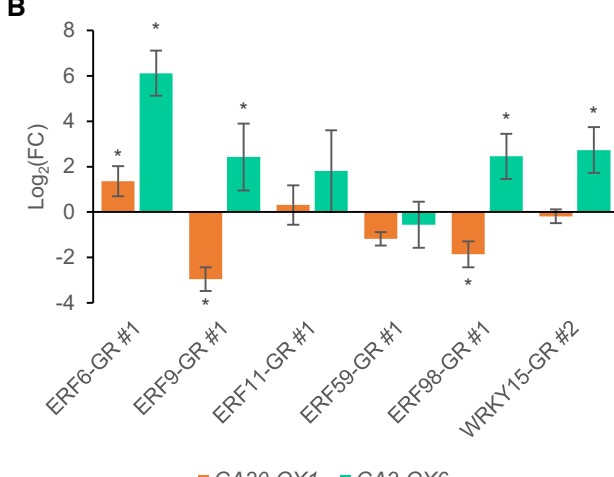

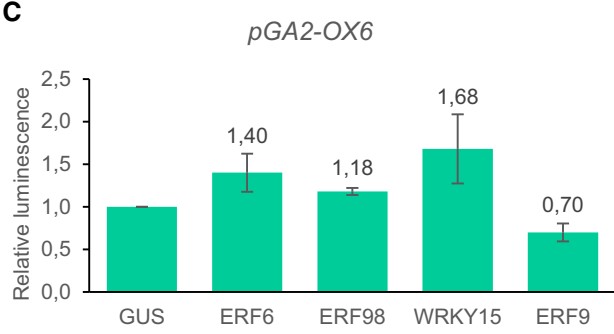

**Figure 5. Expression of four GA biosynthesis and degradation enzymes in wild-type plants upon mannitol treatment and in GOF lines showing an altered growth phenotype in at least two independent lines.**

A  The expression level of two GA degradation and two GA biosynthesis genes in expanding leaf tissue (third leaf at 15 DAS) of wild-type plants 20 min, 40 min, 1 h, 2 h, 4 h, 8 h, 12 h, 16 h, and 24 h after mannitol treatment. The fold changes (FC) were calculated relative to control conditions.

B  The expression of *GA2-OX6* and *GA20-OX1* in expanding leaf tissue (third leaf – 15 DAS), 8 h after transfer to DEX-containing medium to induce overexpression. The FC was calculated relative to control conditions.

C  The effect of ERF6, ERF9, ERF98, and WRKY15 on the *GA2-OX6* promoter determined with transient expression assays. The relative luminescence was calculated relative to the control, 35S::GUS (n = 4 biological repeats).

Data information: In (A), data are presented as mean ± SEM, n = 4 independent experiments, *FDR < 0.05 (unpaired two-sided Student's *t*-test). In (B), data are presented as mean ± SEM, n = 3 independent experiments, *FDR < 0.1 (mixed model analysis, user-defined Wald tests). In (C), data are presented as mean ± SEM, n = 3 independent experiments.

four tested TFs could influence the *GA2-OX6* promoter activity: WRK15 and the two core network members ERF6 and ERF9 (Fig 5C). Interestingly, no significant changes in the expression of GA degradation or biosynthesis genes were observed in the ERF59-GR line (Fig 5B), which showed a significantly larger rosette area when grown under control conditions in the presence of DEX (Fig 4C), suggesting that in this case a GA-independent mechanism might be involved. Taken together, these results show that several TFs of the network have the capacity to affect GA metabolism genes, resulting in rosette

size reduction. Among them, at least two core network members could transregulate *GA2-OX6* expression.

### Perturbing the GRN alters the sensitivity to mild osmotic stress

Because the 20 TFs constituting the GRN were shown to be upregulated upon mild osmotic stress, we subjected the different LOF and GOF lines to a large phenotypic analysis. At 22 DAS, wild-type plants grown under mild osmotic stress conditions (25 mM mannitol) showed an average rosette area reduction of 30% compared to control conditions. Hereafter, all stress-induced size reductions of the analyzed lines were normalized to the reduction in the corresponding wild type, set equal to 1. As such, a mannitol-induced growth reduction of 60% corresponds to a relative reduction of 2. Additionally, we performed leaf series measurements to identify the most affected leaves.

Multiple LOF (*erf2*, *erf8*, *myb51*, *stz,* and *wrky15*) and GOF (ERF2-GR, ERF5-GR, ERF6-GR, ERF9-GR, ERF11-GR, ERF59-GR, and WRKY48-GR) lines showed a significantly altered sensitivity to mannitol (Fig 6A and C; Appendix Figs S3–S22), meaning that the reduction in rosette area upon stress was less (relative reduction < 1) or more (relative reduction > 1) pronounced than in the corresponding wild type. Strikingly, most GOF lines with an altered rosette area were hypersensitive to mannitol (Fig 6C). Regarding ERF9-GR and ERF59-GR, this was visible in two of the three independent GOF lines with a relative reduction of, respectively, 1.64 (*P* = 0.018, Tukey's test) and 2.04 (*P* = 0.014, Tukey's test) and, 1.18 (*P* = 2E-4, Tukey's test) and 1.17 (*P* = 0.017, Tukey's test; Fig 6C). For three TFs (ERF2, ERF5, and ERF11), one of the two independent lines, each time the one with the highest level of overexpression, showed hypersensitivity (Fig 6C; Appendix Figs S4, S5, and S9). For ERF6-GR, the two independent lines apparently showed an opposed sensitivity to mannitol, although this observation should be interpreted with care because seedlings strongly overexpressing *ERF6* are too dwarfed to enable proper growth quantification. The LOF lines *wrky15*, *stz,* and *myb51* were significantly more tolerant to mannitol (relative reduction 0.72, *P* = 8.2E-5, 0.79, *P* = 1.2E-3 and 0.58, *P* = 2.4E-4, Tukey's test, respectively; Fig 6A). Other LOF lines, such as *erf8*, were more affected by the stress treatment than the corresponding wild type (Fig 6A). For most

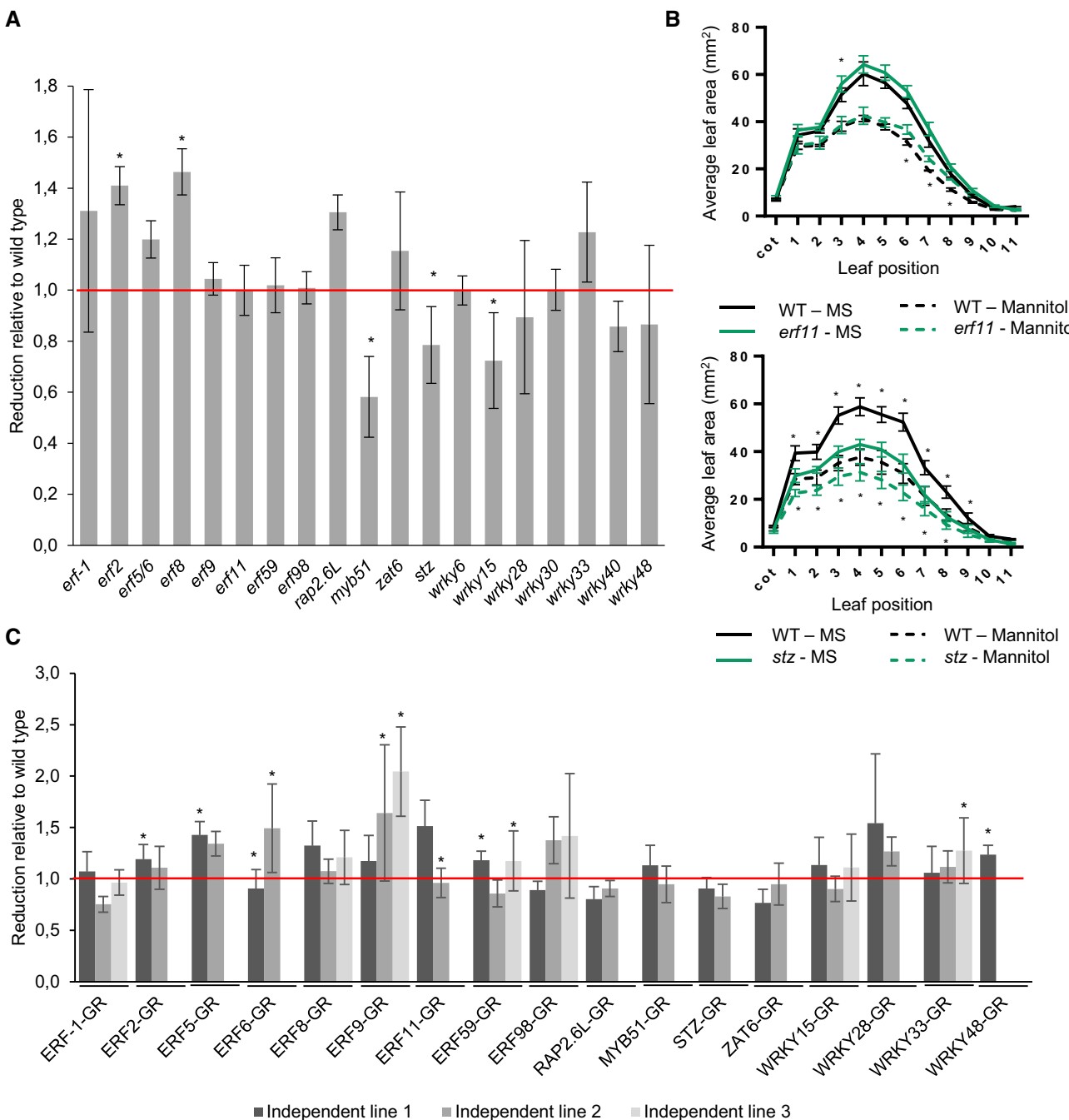

**Figure 6. Phenotypic analysis under mild osmotic stress of loss-of-function (LOF) and gain-of-function (GOF) lines of every TF.**

A  At 22 DAS, leaf series were made of the LOF lines (*n* = 10 plants) grown under control conditions or mild osmotic stress conditions, and the reduction under mild osmotic stress was calculated. The rosette area reduction is presented relative to the corresponding wild type.

B  Leaf series of *stz* and *erf11* mutants grown on control and mild osmotic stress conditions.

C  The rosette area of two or three independent GOF lines (*n* = 10 plants) germinated on DEX-containing control or mild osmotic stress medium was measured at 22 DAS, and the reduction by mild osmotic stress was calculated relative to the control line. "Independent line 1" is the line with the highest overexpression level.

Data information: In (A, C), data are presented as mean ± SEM, *n* = 3 independent experiments, *$P < 0.05$ (Tukey's test). In (B), data are presented as mean ± SEM, *n* = 3 independent experiments, *$P < 0.05$ (mixed model, partial *F*-tests).

mutants, such as *stz,* all leaves had a different size compared to the wild type under mild mannitol conditions (Fig 6B). In contrast, in *erf8* and *myb51*, only the older or younger leaves, respectively, showed a different leaf size compared to the wild type upon mannitol treatment (Appendix Figs S7 and S15). For some lines, such as *erf11*, we could not observe an altered sensitivity on rosette level, whereas on the leaf level, we found differential effects (Fig 6B, Appendix Fig S9). Taken together, these results show that ectopic expression or inactivation of many of the TFs of the GRN affects the sensitivity of plants to mild osmotic stress.

### Complex combinatory regulations add another dimension

To gain insights into the molecular function of the TFs and to study their combined regulatory capacities, we tested the effect of two TFs on a single promoter (Appendix Figs S3–S22). For every promoter, all potential upstream regulators were selected based on the 81 previously identified interactions (Fig 3A) and all pairwise combinations were tested with TEAs.

Five different scenarios were observed when co-transforming two TFs. First, the simultaneous expression of an activator and a repressor could reduce the effect of each individual TF. For example, ERF8 and ERF59 could repress and activate *ERF11*, respectively. When both constructs were co-transformed, pERF11::*fLUC* was less activated than when only ERF59 was present (Fig 7A; Appendix Fig S9C). Second, the regulation of the first TF could be eliminated by the second TF, also when both TFs were activators. For example, ERF98 could activate pERF5::*fLUC*, but when ERF59 or ERF2 were co-transformed with ERF98, the activation was abolished, even though ERF59 and ERF2 are both non-regulating TFs for *ERF5* (i.e., a TF that has by itself no effect on the target promoter) (Fig 7B; Appendix Fig S5C). Third, the effect of one TF could be fully maintained even when expressing it together with a TF with the opposite effect. For example, when ERF8, a strong repressor, was combined with one of the two activators of pRAP2.6L::*fLUC*, ERF98 or WRKY15, pRAP2.6L::*fLUC* remained repressed (Fig 7D; Appendix Fig S12C). Fourth, the intensity of the regulation could be increased by adding a non-regulating TF, suggesting that the non-regulating TF could influence the target gene's expression even if it could not regulate the promoter on its own. For example, ERF6 could activate pMYB51::*fLUC*, whereas ERF98 could not, but the co-transformation of 35S::ERF98 and 35S::ERF6 resulted in a 51% increase in the luminescence signal compared to the single effect of ERF6 (Fig 7C; Appendix Fig S15C). As last, co-transforming two non-regulating TFs could give rise to a regulatory effect. For example, WRKY48 and ERF9 were unable to affect the expression of pERF6::*fLUC*, but the expression of both TFs together did result in the activation of the *ERF6* promoter (Fig 7E; Appendix Fig S6C). For ERF9, in general, the single transformation rendered repression or non-regulation of target genes but when expressed with other non-regulating TFs, activation occurred. This TEA analysis including couples of TFs enabled the establishment of additional regulatory links, and in total, 23 out of the 36 previously unconfirmed interactions from the expression analysis could be confirmed, resulting in a total of 68 confirmed interactions (Appendix Table S4). By evaluating the effect of two TFs on a target gene, we could clearly demonstrate that the network is more complex than initially presented. Some regulations were not visible when considering a one-to-one

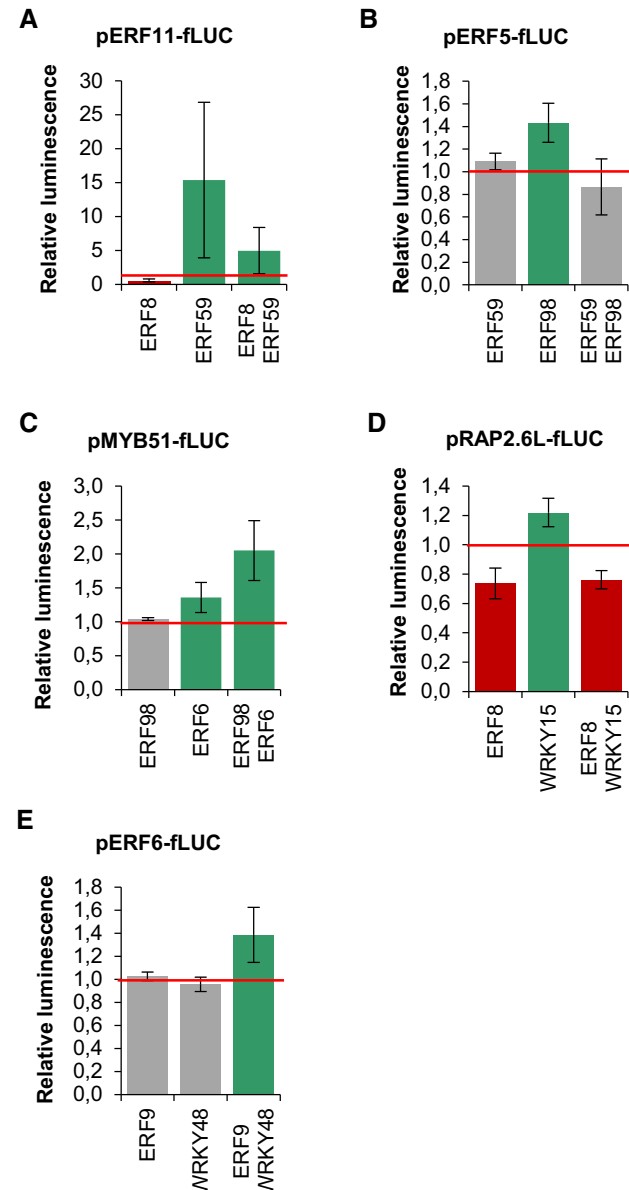

**Figure 7.  Five different effects of two TFs on a common target gene.**

A–E   The effect of the individual and the combination of two TFs on the expression of target genes *ERF11* (A), *ERF5* (B), *MYB51* (C), *RAP2.6L* (D), and *ERF6* (E). The relative luminescence was calculated relative to the control, 35S::GUS (*n* = 4 biological repeats). Green represents activation, red repression, and gray absence of regulation.

Data information: Data are presented as mean ± SEM, *n* = 3 independent experiments.

relation. This extra dimension further increases the complexity of the highly interconnected GRN.

### Co-overexpression of the TFs of the core network leads to diverse but predictable growth phenotypes

To evaluate the combinational effect of two TFs on growth, we crossed the GOF lines of the members of the core network. The GOF lines of all

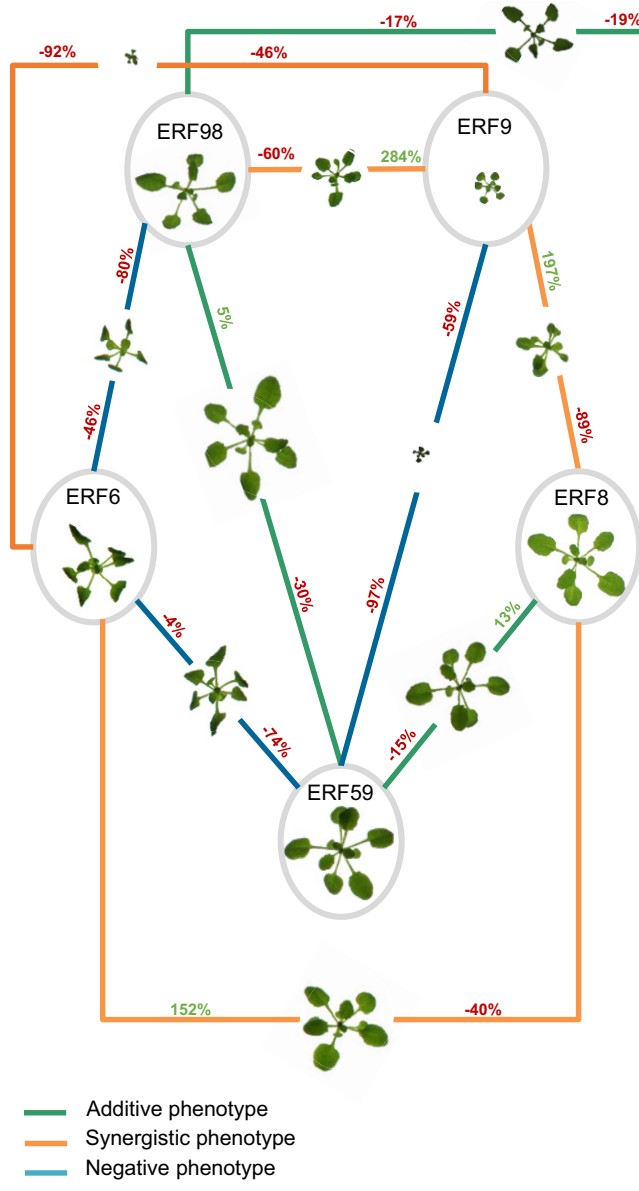

**Figure 8. Phenotypes of the double crosses between core network members.**

The gain-of-function lines of all members of the core network were crossed either with each other, referred to as double cross, or with the control line (GFP-GR), referred to as single cross. The projected rosette area of the F1 double and single crosses and the GFP-GR line, germinated and grown on DEX, was measured at 22 DAS. A representative picture of the single and double crosses is depicted. The values represent the relative increase or decrease in rosette size of the double crosses compared to their parental single crosses. The outcome of every double cross is classified into three groups: additive (green connections), negative (blue connection), and synergistic (orange connections) phenotype. The precise measurements can be found in Fig EV2.

core TFs, except ERF8-GR, showed a growth phenotype: severe rosette size reduction for ERF9-GR and ERF6-GR, moderate reduction for ERF98-GR, and a moderate rosette size increase in the ERF59-GR line (Fig 4). Moreover, they could regulate a large part of the network members and, most importantly, they have common targets, giving us

the opportunity to assess the link between their combined effect on downstream targets and the output, that is, growth.

All pairwise crosses between the GOF lines of the core network members were made, and we refer to them as double crosses (Appendix Table S2). To have comparable overexpression levels with the double cross, every GOF line was also crossed with the control line, referred to as single cross, and used as a control. The projected rosette area of the F1 generation was measured at 22 DAS. To evaluate the genetic interaction between the core transcription factors, the expected rosette area of the double cross, that is, the rosette area when no genetic interaction takes place between the two participating genes, was calculated based on the rosette area of both parental single crosses (Vanhaeren *et al*, 2014) and compared to the observed rosette area. For example, if two single crosses each resulted in a growth reduction of 50%, the double cross is expected to have an even more severe growth reduction of approximately 75% (additive phenotype) if there is no genetic interaction. We could observe three different phenotypes in the double crosses: an expected additive phenotype, an unexpected negative phenotype, or an unexpected synergistic phenotype (Fig EV2). When crossing the growth-promoting ERF59-GR with ERF8-GR or ERF98-GR (ERF59-GRxERF8-GR and ERF59-GRxERF98-GR), an additive phenotype could be observed, which per definition also led to slight compensation of the growth-reducing effect of the other TF (Figs 8 and EV2B). However, the cross between ERF9-GR or ERF6-GR and ERF59-GR (ERF59-GRxERF9-GR and ERF59-GRxERF6-GR) resulted in a negative phenotype. ERF59-GRxERF9-GR showed a severely dwarfed phenotype and was 59% smaller than the already dwarfed ERF9-GR single cross (Figs 8 and EV2C). A similar severely dwarfed phenotype was observed in the ERF9-GRxERF6-GR cross but resulted in a synergistic phenotype; the rosette area of the double cross was 46% smaller than the ERF9-GR single cross and 92% smaller than the ERF6-GR single cross but was still 52% larger than expected (Figs 8 and EV2A). Three other crosses, ERF6-GRxERF8-GR, ERF98-GRxERF9-GR, and ERF8-GRxERF9-GR, resulted in an unexpected synergistic interaction phenotype leading to the partial compensation of the single crosses (Figs 8 and EV2A). In two cases, the cross between a transcriptional activator and a repressor led to partial compensation, indicating common targets (ERF98-GRxERF9-GR), or a transcriptional interaction (ERF6-GRxERF8-GR) (Fig EV2A). To conclude, three of the ten double crosses resulted in an additive phenotype, whereas the other crosses resulted in unexpected phenotypes caused by interactions between the core network members.

## Discussion

### Regulatory redundancy leads to a strongly interwired and robust GRN

It is commonly assumed by biologists that a plant responds to signals via linear pathways: a signal is sensed by a receptor, leading to the activation of a signaling cascade with downstream TFs, which in turn regulate a series of second-order TFs, each responsible for regulating their own output genes. However, in many cases, the sensing of a signal does not result in a single cascade of events (Sasidharan & Mustroph, 2011; Ikeuchi *et al*, 2017; Wang *et al*, 2017), but in the activation of different pathways that are connected

to each other at various levels. A well-known example is the convergence of different pathways at several MAPKs (Xu & Zhang, 2015; Chardin *et al*, 2017). Instead of multiple parallel linear pathways, networks are a more correct view.

In this study, we selected 20 TFs and examined their role in the response to mannitol-induced stress. The 20 TFs could largely regulate each other's expression through 45 confirmed regulatory connections. We discovered that this large amount of connections resulted in redundant regulations, meaning that most TFs have two or more upstream regulators. Only ERF8 and WRKY15 were regulated by one upstream TF, and for ERF9, ERF98, and WRKY28, no upstream regulator was identified. On the other hand, many TFs have only few or even no downstream targets (among the examined genes) because most regulatory connections (39) were originating from five TFs, ERF6, ERF8, ERF9, ERF59, and ERF98, collectively named the core network, and thus, the hubs of the network. Regulatory redundancy within a network leads to a more robust network (MacNeil & Walhout, 2011), which was also observed in the core network by the high overlap in target genes between the two core transcriptional repressors (ERF8 and ERF9; seven target genes) and between the core transcriptional activators (ERF6, ERF59, and ERF98; four target genes between at least two activators). Knocking out ERF6, ERF9, ERF59, or ERF98 of the core network did not result in a growth phenotype, also suggesting redundancy between the core members. We also discovered that the combined regulation of two TFs on the same target gene in our TEAs does not necessarily lead to an additive effect of the single regulations, increasing the complexity even more.

To further explore the co-regulation of a target gene by two TFs at the molecular level, we determined whether there was an overlap in the predicted binding sites in the promoters of shared target genes. With RSAT, a tool specifically designed to detect regulatory signals in non-coding sequences, we could retrieve the exact positions of experimentally determined motifs (known for most TFs but unavailable for ERF9, WRKY6, WRKY28, ZAT6, and MYB51) in the different promoters (Weirauch *et al*, 2014; Medina-Rivera *et al*, 2015; Source Data for Appendix Fig S25). We could observe that the overall effect of the co-regulation of two TFs depended on three factors. (i) The number of DNA-binding motifs: in the promoter of *RAP2.6L,* there is one putative DNA-binding motif for WRKY15 compared to six binding sites for ERF8, suggesting that ERF8 has a stronger connection with the promoter, which was experimentally confirmed (Appendix Fig S25D, Fig 7D, Source Data for Appendix Fig S25). (ii) Competition between two TFs for overlapping DNA-binding motifs: for ERF59, a transcriptional activator, and ERF8, a transcriptional repressor, slightly different but predominantly overlapping sequences are present in the promoter of *ERF11* (Source Data for Appendix Fig S25), leading to an additive effect when both TFs are co-transformed (Appendix Fig S25A, Fig 7A). (iii) The sequestration or recruitment of TFs: in the *ERF5* promoter, the more-abundant motifs for ERF59 overlapped with the few for ERF98, suggesting that ERF59 prevents ERF98 to bind the promoter, which is supported by the loss of increased luminescence signals (Appendix Fig S25B, Fig 7B, Source Data for Appendix Fig S25). Whereas motif analysis provided an indication whether the target promoters could be bound by the upstream TFs, future experiments will need to address to which extend TFs occupy promoters and how this is affected by adverse environmental conditions.

Our network analysis thus clearly shows that presenting signaling cascades as linear pathways is a simplification. The high number of regulatory interactions between these TFs points toward a complex nature of the transcriptional response that ultimately affects growth. However, gene regulatory networks are difficult to study because they are highly complex because of the strongly interconnected wiring, the crosstalk between the individual components and the feedback mechanisms implemented to overcome overactivation and to restrict activity over time (Jaeger *et al*, 2013; Albert *et al*, 2014; Zhong & Ye, 2014). This complexity is needed for the plant to cope with diverse environmental fluctuations and enables fine-tuning.

## An incoherent feed-forward loop enables environmental adaptation

The specific connections between the nodes and edges, that is, the topology of the network, allow the plant to adapt to the environment. For a transcriptional system to enable adaptation, only one of two basic network loops needs to be present: a negative feedback loop or an incoherent feed-forward loop with a delay (Ma *et al*, 2009). A negative feedback loop can be defined as a network motif in which subsequent nodes lead back to the original node, eventually resulting in an inhibitory effect on the original node. An incoherent feed-forward loop can be defined as a network motif in which one pathway inhibits and another activates the output node (Ma *et al*, 2009). For a feed-forward loop to enable adaptation, a delay on the inhibitory pathway must be present. In this way, the activation pathway induces the output node and, after a delay, the inhibitory pathway brings the output node back to its original state. In this GRN, an incoherent feed-forward loop could be found, which could be at the base of the plant's adaptation to the stress signal.

Our expression data have shown that, under standard control conditions, *ERF-1*, *ERF2*, *ERF8*, *WRKY6*, *WRKY15* and *WRKY33* are expressed in expanding leaf tissue (Dataset EV1). We hypothesize that ERF8, a transcriptional repressor that is part of the core network and capable of repressing most network members, suppresses the activation of the network until a stress signal is perceived (Fig 9A). Upon stress perception, the first transcriptional changes occur after 40 min. We found that the expression of the TFs was induced in a sequential manner upon stress, enabling the identification of four different induction groups. The TFs of the first group could directly regulate TFs of a later group, which in turn could regulate their targets, resulting in a cascade of transcriptional regulation. Two activators of the core network, ERF6 and ERF98, are part of the first group and are responsible for the induction of a large part of the network (Fig 9A). For example, ERF98 could directly activate the expression of every gene in the second group, except for *ERF2* (Fig 9A). ERF6 and ERF98 could even act synergistically, as shown for the combined effect of ERF6 and ERF98 on the induction of *MYB51*. Both genes form the first node of the incoherent feed-forward loop and thus the activation pathway (Fig 9B (A)).

On the other hand, *ERF8* and *ERF9*, the genes encoding the two transcriptional repressors are induced 2 h after mannitol treatment. The delayed induction of these repressors is key to the incoherent feed-forward loop (Fig 9B (B)) and likely crucial for the adaptation. The induction of these repressors coincides with the

**Figure 9.   Overview of the transcriptional events following osmotic stress.**

A   We speculate that under normal conditions, ERF8 represses the other network genes. Upon mild osmotic stress (indicated by a red arrow), some genes of the network can be phosphorylated (PTM), a hypothesis based on the literature and the abundance under normal conditions. Subsequently, the expression of the network genes increases during four groups of transcriptional induction. The direct transcriptional regulations of the core network members (ERF6, ERF8, ERF9, ERF59, and ERF98) are depicted with green and red arrows, representing activation and repression, respectively. The regulatory connections of the core network members that were identified when evaluating the effect of two TFs together on a shared target gene are depicted with dashed green and red arrows, representing activation and repression, respectively. The latter regulatory interactions occur in the presence of a necessary transcriptional partner. The color of the nodes represents the strength of the induction. FC = fold change.

B   Schematic representation of the feed-forward loop the network is composed of. Upon input, such as mild osmotic stress, activators of the core network are induced (A) and activates downstream TFs (C). These TFs or another unknown component could induce the expression of the repressors of the core network (B), which in turn leads to the repression of the downstream TFs (C), restoring the original state of the network.

time point at which the expression of most TFs has reached a maximum and subsequently started to decline (Fig 9B (C)). The two repressors are likely responsible for this decline to balance out the network, reducing the strong induction of the first transcriptional group (Fig 9A). In this context, ERF8 is probably responsible for most repressing activities, because the regulatory capacities of ERF9 result in activation in the presence of another activator or non-regulator, as was shown in the TEAs.

In the first 4 h after mannitol treatment, most transcriptional changes occurred. We could speculate that an intermediate steady state is formed at this moment, with most TFs being in balance, leading to the growth inhibition observed after 24 h. This occurs through, for example, the regulation of the expression of *GA2-OX6*, which inactivates GA, resulting in the stabilization of DELLA proteins and growth retardation. Subtler transcriptional changes could still be observed after 24 h and 48 h of stress, potentially regulating the long-term stress response.

**Regulation by two TFs prevents stochastic activation of the GRN**

In addition to the 45 confirmed regulatory interactions, another 23 regulations could be corroborated when co-transforming two TFs, adding to in total 68 confirmed interactions and leading to the observation that for the induction of some downstream TFs, a set of TFs is necessary. For example, for the activation of *WRKY15*, present in the third transcriptional group, at least two of four of its regulators (ERF-1, ERF98, ERF6, and STZ) from the first and/or second group need to be present. We show that at least three TFs (ERF6, ERF9, and WRKY15) could transactivate the downstream *GA2-OX6* gene, which likely leads to growth inhibition upon mild stress. The occurrence of multiple TFs for the regulation of one gene has the important function to decrease stochastic fluctuations in gene regulation and to lower the noise (Swift & Coruzzi, 2017). This means that the chance to activate a stress response target gene is considerably smaller when two upstream TFs control its expression simultaneously rather than only one TF (Swift & Coruzzi, 2017). This safety mechanism thus prevents the random activation of the stress network, which would be detrimental to the plant.

**The mannitol-induced GRN might be part of a central hub in a range of stress responses**

To assess the broader function of the 20 TFs in different stress responses, we explored their transcriptional induction in four previously published datasets, evaluating in total eight different stresses (Table EV3). Almost all TFs (18/20) were significantly upregulated by *Botrytis cinerea* in leaves (Windram *et al*, 2012). In another study, 18, 15, 14, and 12 of the 18 TFs tested by microarray were significantly up- or downregulated in plants 2 h after high salinity, dehydration, abscisic acid, and cold treatment, respectively (Matsui *et al*, 2008). Furthermore, exposure of soil-grown plants to mild drought significantly altered the expression of 17 TFs in growing leaf tissue (Dubois *et al*, 2017). In a fourth study, expression analysis on whole plants exposed to methyl-jasmonate or to the pathogen *Alternaria brassicicola* revealed differential expression of 12 and 14 of the 17 TFs, respectively (McGrath *et al*, 2005). Because the 20 TFs are differentially

regulated in a wide range of stress-related datasets, we speculate that these genes, or at least part of them, function as a hub for the cross talk between different input signals and the downstream effector genes. However, the intensity and the timing of the induction likely depend on the input signal. The dynamics of the network thus change upon different stresses, resulting in different possible output signals and plant responses.

In addition to their differential expression upon other abiotic or biotic stresses, multiple studies also investigated the precise role of several TFs in diverse stress responses. Transgenic lines of multiple TFs showed a sensitivity phenotype upon several stresses, which also points to a broader function of the mannitol-induced GRN upon stress. Under biotic stress, WRKY28 and WRKY48 have been reported to play a role in response to biotrophic pathogens (Xing *et al*, 2008; van Verk *et al*, 2011), whereas ERF9 is a negative regulator of the defense against the necrotrophic fungi *Botrytis cinerea* (Maruyama *et al*, 2013). Several TFs also have well-defined roles under drought or other abiotic stresses. A mutant in *STZ* has previously been shown to be more tolerant to sorbitol-induced stress and salt stress (Mittler *et al*, 2006), which also corresponds to its mannitol-tolerant phenotype observed in this study. In contrast, *erf98* has been described to be more sensitive to salt stress, which we could not observe under mannitol stress (Zhang *et al*, 2012). Overexpression of *WRKY33* has been reported to infer an increased tolerance to high concentrations of salt (Jiang & Deyholos, 2009), whereas inducible activation of *WRKY33* leads to a higher sensitivity to mild mannitol-induced stress. Whereas salt stress is often seen as an osmotic stress, like mannitol, the latter two examples support the previously made observation that salt and mannitol stress responses clearly involve different molecular factors (Claeys *et al*, 2014b). Under mild drought, in soil-grown *erf2* and *erf8* mutants show an increased sensitivity (Dubois *et al*, 2017). The same trend could be observed in this study on low concentrations of mannitol, highlighting a common role in both stress responses. Because the TFs are differentially expressed and mutant lines of the TFs show a sensitivity phenotype upon a range of stresses, we speculate that multiple stress pathways converge in a set of TFs, enabling different outputs.

**Crosses with the GOF lines of the core TFs point out their complex growth-regulating function**

In accordance with the important role that the core network, composed of ERF6, ERF8, ERF9, ERF59, and ERF98, plays in regulating the GRN, all core network genes lead to a growth phenotype when overexpressed or knocked down. Four of them are growth repressors: ERF6, ERF9, and ERF98 caused smaller plants when overexpressed and *erf8* showed an enhanced growth. ERF59 is a growth enhancer (Table EV2).

In addition, we observed pronounced additive, negative, and synergistic effects in crosses between the GOF lines of the core TFs. For example, when combining ERF98-GR with ERF9-GR, the severe growth reduction for ERF9-GR was partially abolished. ERF98 and ERF9 are both situated upstream of the network and could activate most of the network members, ERF98 on its own and ERF9 in combination with other activating or non-regulating TFs, leading to the induction of the stress response and growth inhibition. We hypothesize that when combined, both genes compete for the same target genes, resulting in a less strong

activation of the network and a partial compensation. Other compensation phenotypes when combining two core network TFs could be explained by the sequential regulation of one TF by the other. We could observe growth compensation when combining the repressor ERF8 and activator ERF6 and when combining the two repressors ERF8 and ERF9, possibly as a result of the inhibitory potential of ERF8 on the *ERF6* promoter or of ERF9 on the *ERF8* promoter, respectively. Three other gene combinations, ERF59 × ERF9, ERF59 × ERF6, and ERF98 × ERF6, resulted in negative phenotypes, meaning that the cross has a smaller rosette than expected. Overexpression of both *ERF59-GR* and *ERF9-GR* abolished the growth-promoting function of ERF59 and the dwarfism caused by ERF9 was even more pronounced. We hypothesize that the three combinations resulted in an even stronger activation of the stress-responsive network, because ERF6, ERF59, and ERF98 are strong activators of the network with overlapping but also distinct targets, and ERF9 has, as shown in our TEAs, activation capacities in combination with another TF. It is clear that growth in general is a delicate balance and that the overexpression of one of the core network members can quickly disturb this balance favoring growth inhibition. In order to positively stimulate growth in a more stable way, future research should focus on eliminating the negative growth regulators instead of enhancing positive growth regulators.

### The growth-regulating function of multiple TFs of the GRN is linked to the GA/DELLA pathway

Disruption of the expression of eight of the 20 TFs resulted in a growth phenotype under control conditions, supporting the hypothesis that the network has a pivotal role in regulating growth. We could identify five growth repressors (ERF6, ERF8, ERF9, ERF11, and ERF98) and five growth enhancers (ERF2, ERF59, RAP2.6L, STZ, and WRKY6) (Table EV2). Leaf growth under control conditions has previously been quantified for mutant lines of ERF2, ERF5, ERF6, ERF8, ERF11, RAP2.6L, and WRKY15 (Vanderauwera *et al*, 2012; Dubois *et al*, 2013, 2015, 2017; Zhou *et al*, 2016). RAP2.6L has, for example, been shown to play a role in the regulation of the division of pith cells (Asahina *et al*, 2011), but no clear growth phenotypes have previously been observed in *rap2.6L* mutants or *RAP2.6L* overexpression lines (Krishnaswamy *et al*, 2011; Liu *et al*, 2012). However, when we examined growth in more detail in this study, we could observe that the younger leaves of *rap2.6L* had a reduced size, possibly through decreased cell division. The overexpression of *ERF11* caused a decreased rosette size in this and a previous study (Dubois *et al*, 2015), suggesting that ERF11 is a negative regulator of growth, whereas another study reported that the overexpression of *ERF11* resulted in the promotion of internode elongation (Zhou *et al*, 2016). Even though a contrasting phenotype was observed, a link between ERF11 and the GA2-OX/DELLA pathway was established. *ERF11* overexpression in young leaves resulted in an increased *GA2-OX6* and a decreased *GA20-OX1* expression, whereas *ERF11* overexpression in the internodes resulted in the opposite trend (Zhou *et al*, 2016). Thus, depending on the tissue, the molecular and phenotypic output could be different, but the GA2-OX/DELLA pathway seems central in both cases. More generally, a correlation between a growth phenotype and an alternation in the GA2-OX/DELLA pathway was also observed in this study:

lines overexpressing *ERF6, ERF9, ERF11, ERF98,* and *WRKY15* had a reduced rosette area, a high induction of the network, and an increased *GA2-OX6* and/or decreased *GA20-OX1* expression. For WRKY15 and ERF59, the GOF lines in this study and their previously studied overexpression lines (under the control of the 35S promoter) (Pré *et al*, 2008) showed a contrasting phenotype. In both cases, it is likely that the overexpression level is crucial and we could hypothesize that the growth-promoting or growth-repressing function of the TF depends on an expression optimum.

In conclusion, the highly interconnected gene regulatory network detailed in this study enables to adapt plant growth to osmotic stress and is likely of pivotal importance to regulate growth in response to a wide range of biotic and abiotic cues. The topology of the GRN, including an incoherent feed-forward loop, enables adaptation to mannitol-induced stress, with the efficiency of the GRN in quickly adapting growth to a changing environment being ensured by a strong and fast induction and its robustness by regulatory redundancy.

## Materials and Methods

### Plant material and growth conditions

SALK_036267 [*erf-1*], FLAG_314D04 [*erf2*], SALK_076967 [*erf5*], SALK_030723 [*erf6*], FLAG_157D10 [*erf8*], SALK_043407 [*erf9*], SALK_116053 [*erf11*], GABI_061A12 [*erf59*], SAIL_213_E01 [*erf98*], SALK_051006 [*rap2.6L*], GABI_228B12 [*myb51*], SALK_054092 [*stz*], SALK_061991 [*zat6*], SALK_012997 [*wrky6*], SAIL_1211_H06 [*wrky15*], SALK_092786 [*wrky28*], SAIL_163_A12 [*wrky30*], SALK_006603 [*wrky33*], CSHL_ET5883 [*wrky40*], and SALK_066438 [*wrky48*] were obtained from the SALK collection. The *erf2, erf5, erf6, erf8, erf9, erf11, erf59, erf98, rap2.6L, myb51, stz, zat6, wrky6, wrky33, wrky40, wrky48* mutants have already been described (Appendix Table S1). Mutant lines were genotyped and upscaled simultaneously with the corresponding Col-0 (SALK, GABI) or Ws (FLAG) wild-type line.

To generate the 35S::TF-GR lines, the coding sequence of the transcription factor without STOP-codon, a glucocorticoid domain (GR), and the constitutive 35S-promoter were cloned in pDONR221, pDONRP2RP3, and pDONRP4P1R, respectively, with Gateway Cloning®. A multisite LR recombination was performed to combine the entry vectors into the destination vector pK8m34GW-FAST. Both entry vectors and expression vector were confirmed with sequencing.

Plants were grown *in vitro* at 21°C under a 16-h-day (110 mmol/ (m$^2$s)) and 8-h-night regime on solid 1/2 MS medium (Murashige and Skoog, 1962) containing 1% sucrose.

### Phenotypic and cellular analysis

Four lines were grown side by side on a 14-cm-diameter Petri dish. Per biological repeat, 48 seeds of every line were sown over 12 plates. Half of the plants were grown on solid (9 g/l agar, Sigma) 1/2 MS control medium and the other half on solid 1/2 MS medium with the addition of 25 mM D-mannitol (Sigma). For the GR lines, 5 μM dexamethasone (Sigma) was added to the growth medium. During growth, the plates were randomized. The different lines were

always grown together on one plate with the appropriate control line, Col-0 or Ws wild-type for T-DNA insertion mutants and 35S::GFP-GR for the GR lines. At 22 DAS, a picture of 24 plants per genotype per condition was taken and the projected rosette area was measured using the software program ImageJ version 1.45 (National Institutes of Health; http://rsb.info.nih.gov/ij/). Ten plants were harvested and leaf series were made by which each individual leaf was cut and laid out from old to young on a square agar plate. Plates were photographed and pictures were analyzed using ImageJ to measure the size of each individual leaf. The third leaf was harvested from the plates and cleared in 100% ethanol. After clearing, the ethanol was replaced with lactic acid and the leaves were mounted on microscopic slides. The leaves were photographed with a binocular, and the three leaves with an area closest to the median were used for cellular analysis. Per leaf, approximately 100 abaxial epidermal cells were drawn with a DMLB microscope (Leica) fitted with a drawing tube and a differential interference contrast objective. Pictures of the cell drawings were used to measure the average pavement cell area and number with ImageJ as described before (Dubois *et al*, 2015).

For the statistical analysis of the rosette area, an ANOVA was performed for each line separately in R (version 3.3.2) (http://www.R-project.org/) (R Core Team, 2015). Fixed factors in the model were line, treatment, and their interaction term. Each experiment was repeated three times, and the factor repeat was included as a block effect. Model building was used to achieve the best model.

A linear mixed model was fitted to the leaf area data. The data are clustered as measurements were done on leaves originating from the same plant. For the LOF and GOF lines grown on 1/2 MS medium, model building started with a saturated mean model containing the main effects of genotype and leaf and the interaction term. The Kenward–Roger approximation for computing the denominator degrees of freedom for the tests of fixed effects was used. Several structures were tested for the variance–covariance matrix: unstructured, (heterogenous) compound symmetry, (heterogenous) autoregressive, and (heterogenous) banded Toeplitz. Based on the AIC values, an autoregressive structure was assumed. A random effect for repeat was included in the model to account for the correlation between plants grown at the same time. The main interest was in the comparison of each line with the wild type at the different leaves. Type III tests of fixed effects were calculated to verify that there was a significant interaction term at the 0.05 significance level. Simple *F*-tests of effect for genotype were carried out at each leaf. For those leaves showing a significant *F*-test ($P < 0.05$), pairwise comparisons were estimated between genotype and wild type. At each leaf, correction for multiple testing was done applying the Dunnett method. The analysis was performed with the mixed and plm procedure of SAS (Version 9.4 of the SAS System for Windows 7 64 bit, Copyright © 2002–2012 SAS Institute Inc. Cary, NC, USA [www.sas.com]). Residual diagnostics were carefully examined.

### Phenotypic analysis of the crosses

The 35S::TF-GR lines were crossed with each other (double cross) and with a 35S::GFP-GR line (single cross). The F1 generation was used to perform the phenotypic analysis. Six or seven lines were grown on a 23-cm square plate. Per biological repeat, eight to nine

seeds per line were sown on solid (9 g/l agar, Sigma) 1/2 MS medium with the addition of 5 μM dexamethasone (Sigma). During growth, the plates were randomized. The different lines were grown together on one plate with the appropriate control lines, 35S::GFP-GR or the single crosses. At 22 DAS, a picture of the square plates was taken and the projected rosette area was measured using the software program ImageJ version 1.45 (National Institutes of Health; http://rsb.info.nih.gov/ij/).

To estimate the interaction between two genes, following parameterization was used. Two dummy variables were created, P1 and P2. P1 is equal to 1 when the line contains gene 1 and 0 otherwise. P2 is equal to 1 when the line contains gene 2 and 0 otherwise. To stabilize the variance, a log2 transformation was performed on the rosette area data. The model contained the dummy variables P1, P2, and the interaction term. With this parameterization, the regression coefficient of the interaction term has the direct interpretation of deviation from additivity. Additivity corresponds to the hypothesis (Vanhaeren *et al*, 2014):

$$E\{log2(RAdc)\} = E\{log2(RAsc1)\} + E\{log2(RAsc2)\} - E\{log2(ref)\}$$

The interaction was said to be synergistic or negative when the regression coefficient of the interaction term was positive or negative, respectively, and with a $P < 0.05$.

### Expression analysis

For the expression analysis performed on wild-type plants (as shown in Appendix Fig S1, Figs 1, 2 and 5), a 14-cm-diameter Petri dish with solid 1/2 MS medium (6.5 g/l agar, Sigma) was overlaid with a nylon mesh (Prosep) of 20-μm pore size. On each plate, 64 (for harvest on 9 DAS) or 32 (for harvest on 15 DAS of 22 DAS) wild-type seeds were sown and during the growth plates were randomized. Half of the plants were transferred to control 1/2 MS medium, the other half to 1/2 MS medium containing 25 mM mannitol to induce mild osmotic stress. The transfer was enabled by picking up the mesh with the plants on top and laying it out on a fresh plate. For the plants transferred at 9 DAS and 22 DAS, the third leaf was harvested 24 h after transfer. For plants transferred at 15 DAS, the third leaf was harvested 20 min, 40 min, 1 h, 2 h, 4 h, 8 h, 12 h, 16 h, 24 h, and 48 h after transfer (one plate per time point was harvested).

For the expression analysis of the GR lines (as shown in Figs 3 and 5), four lines were sown side by side on a 14-cm-diameter Petri dish with solid 1/2 MS medium (6.5 g/l agar, Sigma). Each plate contained the appropriate control 35S::GFP-GR and was overlaid with a nylon mesh (Prosep) of 20-μm pore size. In total, 120–160 seeds of every genotype were sown. At 15 DAS, the plants were transferred to solid 1/2 MS medium containing 5 μM DEX to induce the overexpression. The third leaf was harvested from 12 to 16 plants per genotype 1, 2, 4, 8, and 24 h after transfer.

The samples harvested on 15 DAS and 22 DAS were immediately frozen in liquid nitrogen. The 9 DAS samples were harvested on RNA-stabilizing solution RNAlater (Ambion), according to the manufacturer's instructions, and subsequently, the third leaf was microdissected and frozen in liquid nitrogen. All samples were ground with a Retsch machine and 3-mm metal beads. Subsequently, RNA was extracted with TriZol (Invitrogen) and further purified with the

RNeasy plant mini kit (Qiagen). An amount of 100 ng with 5 μl extra for quality control was sent to the Nucleomics Core in Leuven for an nCounter Nanostring® analysis (http://www.nanostring.com/applications/technology). Five housekeeping genes (AT1G13320, AT2G32170, AT2G28390, AT5G15710, AT4G24550) were included to normalize the data. Two sequence-specific probes were designed and synthesized at Nanostring Technologies (Seattle, USA) for 36 genes (including five housekeeping genes, Dataset EV1 and Source Data for Fig 3). One probe was used to hybridize and immobilize the complementary RNA molecules. The other probe was used to detect the individual RNA molecules. The induced overexpression was confirmed for all GOF lines in the Nanostring experiment, except for the ZAT6-GR line.

For the statistical analysis of the expression analysis in wild-type plants, Student's $t$-tests were performed on $\log_2$-normalized values for each gene separately. The retrieved $P$-values were adjusted for multiple testing with the FDR method in R (version 3.3.2).

For the statistical analysis of the expression analysis in the GR lines, a mixed model analysis was performed on $\log_2$-normalized values for each experiment and each gene separately using the mixed procedure in SAS (Version 9.4 of the SAS System for Windows 7 64 bit. Copyright © 2002–2012 SAS Institute Inc. Cary, NC, USA [www.sas.com]). Fixed factors in the model were line and time and their interaction term. Each experiment was repeated three times, and the factor repeat was put as random effect in the model to account for correlations between data observed within the same repeat. Denominator degrees of freedom were calculated using Satterthwaite's approximation as implemented in SAS. The contrasts of estimate were the differences between each line and the reference line at each time point. All $P$-values from one experiment were adjusted for multiple testing with the FDR method as implemented in the multitest procedure from SAS (Benjamini & Hochberg, 1995).

For the initial selection of the TFs (as shown in Appendix Fig S1), the expression level of OBP1, MEE3, AT5G58900, and WRKY18 was measured at 15 DAS with qRT-PCR. To confirm the overexpression of ZAT6 in the 35S::ZAT6-GR line, the ZAT6 expression was measured in the previously described samples with qRT-PCR. For cDNA synthesis, the iScript cDNASynthesis Kit (Bio-Rad) was used using 1,000 ng of RNA as starting material. qRT-PCR was performed with the LightCycler 480 Real-Time SYBR Green PCR System (Roche). The data were normalized against the average of housekeeping genes AT1G13320 and AT2G28390, as followed: $dC_t = C_t$ (gene) $- C_t$ (average [housekeeping genes]) and $ddC_t = dC_t$ (Control) $- dC_t$ (Treatment). $C_t$ represents the number of cycles at which the SYBR Green fluorescence reached a threshold during the exponential phase of amplification. Primers were designed with the QuantPrime web site (http://www.quantprime.de/) and are as followed: OBP1 (AT3G50410), TCAGCTTTGGACTCGGAAGAGC, and TCGTCGTTGTCGCAGTACCAAC; MEE3 (AT2G21650), TCACG TGCCATTCCCTGACTAC, and TGCAGCTTCATGCTTCTCATCCTC; MYBx (AT5G58900), CTTGGACGGAGGAAGAACACAAGC, and TTG TGTTGGCGTTCGCGTTATC; WRKY18 (AT4G31800), TGGACGGTT CTTCGTTTCTCGAC, and TCGTAACTCACTTGCGCTCTCG; ZAT6 (AT5G04340), TCTACAAGCCACGTCAGCAGTG, and TTCCGGTATC GGCGGTATGTTG; Housekeeping gene 1 (AT1G13320), TTGGTGCT CAGATGAGGGAGAG and TTCACCAGCTGAAAGTCGCTTAG; Housekeeping gene 2 (AT2G28390), CAAGGCAGGAAATCACCAGGTTG and CTGTACAGCTGATGCAGACCAG.

## Transient expression assay

The transient expression assays were performed as previously described (Vanden Bossche et al, 2013). The 35S::TF (p2GW7) and pTF::fLUC (pm42GW7) constructs were generated using the Gateway cloning system and a concentration of 2 μg was used. When combining two effector plasmids (35S::TF), 2 μg of each plasmid was added. The co-transformation of the individual effector plasmids with the reporter plasmid was included as control. Therefore, 2 μg of the effector plasmid, 2 μg of 35S::GUS, and 2 μg of the reporter plasmid were added in order to have the same relative amount of plasmid. To enable the translocation of TF-GR fusion proteins to the nucleus, a 10 μM dexamethasone treatment of 4 h was performed prior to the lysis of the protoplasts.

## Visualization of the network

For the visualization of the network, the software Cytoscape (version 3.2.1) was used (Lopes et al, 2010; Smoot et al, 2011). A text file was compiled in which each row corresponds to one interaction. The file contained four values for each row: the tested regulator (the 35S::TF-GR line in Fig 3A or the 35S::TF in Fig 3B), the target gene (the differentially expressed gene in Fig 3C or the pTF::fLUC in Fig 3B), the expression value (the estimated $\log_2[FC]$ of the expression analysis in Fig 3A or relative luminescence of the TEA experiments in Fig 3B), and the FDR value of the 35S::TF-GR expression analysis. A network was generated from the imported text file in which the 35S::TF-GR lines or the 35::TFs from the TEAs were defined as the source interaction, the differentially expressed genes, or as target interaction and the interaction type as pd (protein–DNA interaction).

## Motif analysis

The position-specific scoring matrices (PSSMs) from the DNA-binding motifs of the TFs were retrieved from the CIS-BP database (http://cisbp.ccbr.utoronto.ca/; Weirauch et al, 2014). For five genes, ERF9, WRKY6, WRKY28, ZAT6, and MYB51, the binding motif was not yet experimentally determined and was left out of the analysis. The PSSMs were converted from cis-bp format to transfac format with the convert matrix tool on RSAT (http://floresta.eead.c sic.es/rsat/; Medina-Rivera et al, 2015). To retrieve the exact positions and significance of motifs in the different promoters, the pattern-matching tool in RSAT was used (Source Data for Appendix Fig S25). A matrix scan (full options) with an organism-specific background model (Arabidopsis thaliana), no masking, and a $P$-value threshold of 0.001 was used to search for individual matches. All other settings were kept on default.

## Data and software availability

The datasets produced in this study are available in the following databases:

- nCounter nanostring experiments in wild-type plants: ArrayExpress, E-MTAB-6205 (https://www.ebi.ac.uk/arrayexpress/experiments/E-MTAB-6205/).

- nCounter nanostring experiments in 17 GOF lines: ArrayExpress, E-MTAB-6209 (https://www.ebi.ac.uk/arrayexpress/experiments/E-MTAB-6209).

**Expanded View** for this article is available online.

## Acknowledgements

We thank the Systems Biology of Yield group for all the helpful discussions and the inspiring environment, Katrien Maleux, Twiggy Vandaele, and Liesbeth De Milde for the practical help, Nico Smet, Thomas Farla, Miguel Riobello y Barea, Wilson Ardiles-Diaz, and Carina Braeckman for all the technical support, and Alexandra Baekelandt, Hannes Claeys, Hannes Vanhaeren, Annick Bleys, and Nathalie Gonzalez for help in improving the article. This research received funding from the Bijzonder Onderzoeksfonds Methusalem Project (BOF08/01M00408). L.V.d.B is a predoctoral fellow of the FWO SB (project no. 131013). [Bijzonder Onderzoeksfonds (BOF); Fonds Wetenschappelijk Onderzoek (FWO)]

## Author contributions

LVB, MD, and DI designed the research; LVB and MV performed the experiments; LVB analyzed the data; LVB, MD, MV, and MM generated plant material; VS and LVB performed the statistical analysis; LVB, MD, and DI wrote the article.

## Conflict of interest

The authors declare that they have no conflict of interest.

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
