## [Review Process File · Molecular Systems Biology]

From network to phenotype: the dynamic wiring of an Arabidopsis transcriptional network induced by osmotic stress

Lisa Van den Broeck, Marieke Dubois, Mattias Vermeersch, Veronique Storme, Minami Matsui and Dirk Inzé

Review timeline:

Submission date:	23 June 2017
Editorial Decision:	1 August 2017
Revision received:	31 October 2017
Editorial Decision:	18 November 2017
Revision received:	25 November 2017
Accepted:	29 November 2017

Editor: Thomas Lemberger

Transaction Report:

1st Editorial Decision

1 August 2017

Thank you again for submitting your work to Molecular Systems Biology. We have now heard back from the two referees who agreed to evaluate your manuscript. As you will see from the reports below, the referees find the topic of your study of potential interest. They raise, however, substantial concerns on your work, which should be convincingly addressed in a major revision.

Without repeating all the points raised in the reports below, the major issues refer to the following:

- variability across transgenic lines, both at the phenotypic or expression level, seems to be an issue that affect the conclusiveness of some of the experimental series reported in the paper.
- the limitations of some of the assays, for example when involving overexpression, should be clearly acknowledged and the interpretation of the results should be toned down as appropriate or formulated in a more rigorous manner.

On a more editorial level, the manuscript would benefit from delineating clearer conclusions that are accessible to a broad audience and non-specialists who are not used to 'systems biology jargon'. We would strongly encourage you to revised the text and the abstract in this direction.

We would also kind ask you to deposit datasets in public repositories (see <http://msb.embopress.org/authorguide#datadeposition>) when available or as "Expanded View Dataset" files to be uploaded to the tracking system. Please include a formal structured "Data Availability Section" after the Materials & Method section, based on the following model:

#Data and software availability

The datasets and computer code produced in this study are available in the following databases:

- [data type]: [full name of the resource] [accession number/identifier] ([doi or URL or identifiers.org/DATABASE:ACCESSION])
- RNA-Seq data: Gene Expression Omnibus GSE46843
[<https://www.ncbi.nlm.nih.gov/geo/query/acc.cgi?acc=GSE46843>]
- Chip-Seq data: Gene Expression Omnibus GSE46748
[<https://www.ncbi.nlm.nih.gov/geo/query/acc.cgi?acc=GSE46748>]
- Protein interaction AP-MS data: PRIDE PXD000208
[<http://www.ebi.ac.uk/pride/archive/projects/PXD000208>]
- Imaging dataset: Image Data Resource doi:10.17867/10000101
[<http://doi.org/10.17867/10000101>]
- Modeling computer scripts: GitHub
[<https://github.com/SysBioChalmers/GECKO/releases/tag/v1.0>]

REVIEWER REPORTS

Reviewer #1:

The dynamics and architecture of the gene regulatory networks that regulate plant responses to environmental stress are largely unknown. This work characterizes the response of 20 transcription factors in *Arabidopsis thaliana* that are upregulated during osmotic stress. By measuring gene expression at different times and in different tissues after induction of osmotic stress, the authors captured dynamic changes in gene expression that demonstrated that the majority of these TFs were induced within 2 hours in growing, but not mature leaves. Inducible overexpression lines and transient expression assays revealed that there were 68 direct interactions between these 20 TFs. A group of five "core" ERF genes (ERF6, 8, 9, 59, and 98) were responsible for most of these interactions. The authors tested loss and gain of function lines to identify the phenotypic roles of individual TFs, as well as pairs of TFs. They found that a pair of TFs could act synergistically or repressively (could either have additive effects or could repress the function of another TF). This work highlights how spatial and temporal dynamics of GRNs are critical for understanding how complex interactions control physiological responses. Although the genes described here have previously been linked to the stress response, the major significance of this work is the presentation of a detailed understanding of how these TFs interact to form a complex and novel network of TFs capable of responding to changing environmental conditions.

The key conclusions of the study seem to be consistent with the data presented, except for the following points.

- 1) On page 7, the authors state that some of the gene expression data demonstrate an oscillation pattern, which the authors suggest indicates the presence of indirect effects. However, using cyclohexamide to inhibit protein synthesis would be a better way of assessing indirect interactions.
- 2) In Figure 3 (described on page 8), the authors group their TF induction data into four waves, categorized based on whether the expression of a gene exceeds a log₂ (fold change) threshold of greater than 1 at 40 minutes, 1 hour, 2 hours, or 4 hours. However, in figure 3, it looks like many of the genes in wave 1 and wave 2 have very similar profiles, and that most of the genes in these waves reach their maximum at the same time (around 2 hours). This is also true of some of the genes in wave 3. Categorizing these expression patterns by initial rate or expression maximum would be a more useful way to understand the kinetics of the network.

Minor points:

I think this paper would benefit from making the major findings more accessible in the abstract and conclusions. In particular, I think the following points should be easier to locate in the paper:

- 1) The identities of the 5 core TFs.
- 2) The number of total direct interactions between the 20 TFs identified (I believe this number is 68).

- 3) The number of total direct interactions linked to the 5 core TFs (I could not locate this number).
- 4) The maximum change in growth caused by loss or gain of function lines.

Reviewer #2:

Summary and general comments

This work builds on a series of previous publications where the authors have investigated to response to a relatively mild osmotic stress imposed by exposing seedlings to 25 mM mannitol in agar plates. In this condition the plants have reduced growth but otherwise have essentially normal development. The authors have used to system to identify the regulatory mechanisms by which plants restrict their growth under mild osmotic stress. Understanding such growth regulation mechanisms have a number of implications in plant stress biology. Such growth restriction is an adaptive response for plants in many environments; however, being able to turn off or sidestep the growth regulation mechanisms may be a way to improve productivity of crop plants. As part of their previous studies, this group has conducted extensive transcriptome analysis to find differentially regulated genes in the mild osmotic stress conditions. Here they take a group of transcription factor genes found to be upregulated in response to mild osmotic stress and attempt to construct a gene regulatory network (ie: find which of these transcription factors may activate or repress others in this group). They do this via physiological experiments to examine growth and gene expression of their transcription factors in mutant lines lacking activity of a specific transcription factor or in transgenic lines with ectopic expression (driven by 35S promoter) of an inducible form (fusion to glucocorticoid receptor to control entry into the nucleus) of each transcription factor. They also cross their inducible expression lines together to test for additive or antagonistic interactions between transcription factors in their putative network. In another set of experiments, they use transient expression in protoplasts to assay which of their transcription factors can activity expression of other transcription factors in their group. They compare this data an analysis of cis elements in the gene promoters to make sense of the patterns of activation/repression as well as additive and antagonistic interactions among their set of transcription factors. They find that their set of transcription factors do cross regulate each other and there are both additive and synergistic interactions. Some of these interactions are expected based on previous data or analysis of cis-elements but others are not. Thus, it is possible that the transcription factors they analyzed are part of a gene regulatory network. How things are arranged in this network and whether is is a fairly complete network or whether other transcription factors could also be part of this network (for example, factors whose expression is repressed by 25 mM mannitol) is not as clear.

The strength of this work is its scale and ambition. I am not aware of any work in plant stress biology (or perhaps in all of plant biology) that has attempted a comprehensive test of physiological and regulatory interactions between such a large set of genes. Such a work should be of great interest to those working in plant stress biology, as well as a broader audience interested in transcriptional regulation. I say should be as I think in its present form this paper would disappoint a good portion of its readers. This is because there is no overall conclusion that emerges from this work. Many would want to know how we can use the new knowledge of this gene regulatory network to perturb it toward allowing more growth under mild stress or further restricting growth. There is no conclusion and no discussion of this point. The discussion as it is rambling through several topics while making no conclusion. Also a number of things related the network analysis or prediction (such as the equation on page 14 or the usage of "incoherent feedforward loop" on page 16 and elsewhere) are used without any context or explanation. I think that the authors should think carefully of what they are trying to accomplish with this study and who their main audience is (presumably plant stress biologists) and write a paper that is more clear and useful to this audience and in which the main conclusions are more clear. From a technical standpoint, the study suffers from being broad but shallow (see details below) and this also introduces doubt about what can really be concluded from this work.

Major points:

There are some major technical points that I have concern about. First, in their analysis plants with ectopic expression of transcription factors (Gain of Function-GOF lines in their terminology) they use of only two independent transgenic lines per construct. This problematic as in some cases the two lines give conflicting results. For example in figure 4C where ERF1-GR has one line with increased growth and one line with decreased growth, ERF2-GR has one line with increased growth and one line with no change, WRDK28-GR has one line with decreased growth and line with no

change, WRDK33-GR has one line with increased growth and one line with no change and WRKY48 has only one line shown. Similar discrepancies can be found in Figure 6C where ERF2, ERF5, ERF11, ERF59 and ERF98 all show dissimilar results between the two transgenic lines. So then how to interpret this data as the whether the ectopic expression has an effect? It seems that for their analysis of cell number and cell area in figure 4 E and F (and also presumably for the crosses conducted later in the study) they just pick the one transgenic line that has an effect (I do acknowledge them for clearly labeling which transgenic line was analyzed). But then how to know whether this is a real effect of the transgene rather than a mutation(s) introduced during transformation? It is even more confusing as they do check that the two transgenic lines have similar levels of gene expression. Such a shallow analysis and just picking the one transgenic line that shows you what you want to see would not be allowed in a more targeted study of a few genes and is not sufficient here either. They need to have 3 or more independent transgenic lines per construct to know what the real effects are. I suspect this issue also affects their gene expression results but it is more difficult to parse out whether the data shown are from both transgenic lines or just the one that showed a difference.

Second, they use protoplast transient expression assays (TEA assays in their terminology) to test cross regulation among different transcription factors in their putative network. While this can be a useful assay, it does have limitation and their interpretation of such data stretches beyond the limitations of this assay. These assays force high level expression of the transcription factor and thus can lead to reporter activation by transcription factors that would not normally activate the reporter in planta. Indeed, they themselves say (bottom of page 7) that the results of the stable transgenic lines expressing transcript factor-GR receptor fusions and transient expression assays do not always agree. They make the conclusion that the transient expression assays more likely represent the actual endogenous activity of the transcription factor. This is not true, or not necessarily true, because of the enforced high level expression in the protoplast assays and because protoplasts are already stressed cells prone to different responses than an intact plant.

Also, they also do transient expression experiments where two of the transcription factors are expressed at once. This leads to various kinds of effects which they try to interpret based on looking at the pattern of cis-elements in the promoters of their genes. They even say (last paragraph on page 13) that they use such search approach to "test whether two TF compete for binding sites in the promoter of a shared target gene". This is flawed. Even though these transcription factors come from relatively well characterized families and their core recognition sites are known, their actual promoter binding sites (which can be influenced by sequences flanking the core recognition sites, epigenetic regulation, other promoter binding proteins and other factors) are not known. And, some of the interaction effects they observed could be explained by different protein levels of the two transcription factors (protein levels are not measured). If they want to talk about promoter binding or competition for promoter binding, then they need to actually measure promoter occupancy of their transcriptions factors using intact plants. The whole paragraph at the bottom of page 13 is at best a side point for the discussion as it is speculation and not based on actual data.

Related to this concern, it must be pointed out that their conclusions about direct versus indirect regulation are based mainly on time course data of which factor is induced first, which one a bit later etc. The networks they draw and the edges the populate them with are based only on expression changes. There is no data in this study of who is actually binding to whose promoter. They do acknowledge (bottom of page 6 and page 7) that such connections could be direct or indirect and that their putative network contains many indirect connections. But then they seemingly forget about this distinction in other parts of the manuscript. The transient expression assays are conducted in plant cells so indirect effects are possible-in other words their introduced transcription factors could activate the reporter construct via mechanisms other than directly binding to its promoter. Here again they need to either measure promoter occupancy or more clearly acknowledge the limitations of the assays they use and be appropriated cautious in their conclusions.

It seems that their transgenic plants expressing transcription factor GR-receptor domain fusions were germinated and growth on continuous supply of DEX and the final plant size measured. How to rule out effects of some lines germinating more quickly, or slowly, than others? This also seems to nullify some of the advantages of making transgenic lines with inducible activation of the transcription factor.

It also must be pointed out that no outgroup of transcription factors analyzed. Most of the effects on growth they observed are relatively subtle (the previously studied ERFs which are also included here are an exception). How do we know that similar effects would not be observed just by randomly picking and analyzing members of the ERF or WRKY gene families (or other types of transcription factors)?

Specific comments (some of these are more minor issues, some are specific examples of the concerns expressed above)

1. Page 8: The "GRN is activated in four waves" section seems out of place here and would seem better presented earlier in the manuscript as preliminary data which helps to interpret the other analysis of transgenic plants.
2. Figure 4A: it seems that their analysis of T-DNA mutants (LOF lines in their terminology) includes only one T-DNA allele for each gene. As explained above for the limited number of independent transgenic lines assayed, this would be considered problematic in more targeted studies and must be kept in mind when interpreting this data also. There is no mention in the manuscript of whether the T-DNA mutants and transgenic GOF lines have opposing effects on growth or gene expression as would be expected. For example, the wrky15 T-DNA line has reduced growth (Fig 4A) and the WRKY15-GR overexpression line also has reduced growth (Fig 4C)-how to explain this?
3. Page 11, authors state "To evaluate whether the observed changes in GA2-OX6 expression were a direct or indirect effect of the overexpressed TF, the binding of the TF on the 2-kbp GA2-OX6 promoter was evaluated with TEAs in tobacco protoplasts." This is incorrect, the transient expression experiments do not assay promoter binding (there is no data on promoter binding in this manuscript).
4. Top of page 12: why is this normalization scheme for the growth data used here and not in other experiments such as those presented in Figure 4?
5. Page 14, The authors state: "The expected rosette area of the double cross was calculated based on the rosette area of both parental single crosses according to the following formula: $\log_2(\text{expected double cross}) = \log_2(\text{single cross 1}) + \log_2(\text{single cross 2}) - \log_2(\text{control})$ and compared with the observed area via the chi-square test (Vanhaeren et al, 2014)." How is this formula derived and what is its relevance here? And, it seems the result of this comparison between observed and predicted growth is not actually presented as it is not shown in Figure 8 (note that I cannot find figure EV2 in my review material).
6. Page 16: What is an "incoherent feedforward loop"? The following sentence "When looking at the network globally, an incoherent feedforward loop seems to be at the base of the adaptation mechanism (see below)" adds nothing. As stated above, consider what your audience is and what the main conclusion is and try to present things more clearly (maybe I am not the intended audience).
7. Likewise, having section headings of "topology of the network" and "wiring of the network" is just obfuscates things.
8. Page 16-17, this paragraph about post translational modification is OK as you clearly state that it is speculation; however, the fact that you actually drew phosphorylation into your model in Fig 9 is not as it presenting this as fact when you don't actually have such data.
9. Page 17 authors state " Because the network genes encode relatively small proteins, we speculate that the time between the detection of the mRNA transcript with e.g. qPCR, the translation and folding of the protein and the subsequent induction of the target gene is less than half an hour" This statement is just silly, translational rate and preferentially translational happen and matter a lot more than just the size of the protein.
10. Page 17: "Our TEAs have shown that two TFs could form combinations to regulate their downstream targets." You do not show anything about transcription factors forming combinations. Please be more careful with your wording.
11. Page 18: "45 direct regulatory interactions were experimentally validated with TEAs and when combining two TFs," Please note the above stated concerns about calling these direct interactions in the absence of promoter occupancy data and based solely protoplast transient expression.
12. Page 18: "and is hereafter referred to as the core network." You have referred to the core network many times before this statement. Are you now referring to a different core network?
13. Bottom of page 18: here again "incoherent feedforward loop" with no explanation of what this is or how it is relevant.
14. Page 20-21: this part is even more rambling than the rest of the discussion. What is your main point?
15. Bottom of page 21: "indirect strong repression of WRKY30". Here again, you cannot say for sure what is direct and indirect.
16. Overall, I find the discussion very rambling and not clearly state what the authors view as the main and most novel findings of this study. The discussion should be shortened and restructured.
17. There are cases throughout the paper where there are small grammar errors (tenses and subject

verb disagreement for example or missing verb, incorrect phrases) and editing of the paper would improve clarity.

1st Revision - authors' response

31 October 2017

We appreciate the time you and the reviewers invested in our manuscript and are very grateful for the opportunity to resubmit to *Molecular Systems Biology*.

The major issues that were raised are the following:

- **Variability across transgenic lines, both at the phenotypic or expression level, seems to be an issue that affect the conclusiveness of some of the experimental series reported in the paper.**

We now phenotypically analyzed an additional independent overexpression line for a large number of the transcription factors that are part of the gene regulatory network under study. The analysis allows for a better supported conclusion whether a transcription factor has a growth promoting or repressing activity.

- **The limitations of some of the assays, for example when involving overexpression, should be clearly acknowledged and the interpretation of the results should be toned down as appropriate or formulated in a more rigorous manner.**

We clarified some of our statements and reasoning, adjusted the results and extensively rewrote the discussion (see below). Unfortunately, in the limited timeframe of the rebuttal it was not possible to perform promoter occupancy experiments for all 20 transcription factors. Instead, we added, in the discussion, results of a recently published (Barlett et al., 2017) genome-wide transcription factor – promoter interaction analysis in which a number of the interactions found in this study were detected. These data further validate the robustness of the experimental data presented in the submitted manuscript.

- **On a more editorial level, the manuscript would benefit from delineating clearer conclusions that are accessible to a broad audience and non-specialists who are not used to 'systems biology jargon'. We would strongly encourage you to revised the text and the abstract in this direction.**

To make this manuscript accessible for a broad audience, we rephrased some jargon and defined the crucial systems biology terms. We rewrote the abstract and discussion and rephrased the section headings to bring our conclusion more forward (see below).

A detailed reply to each reviewer comment is provided below. We hope you will find this manuscript to be improved and acceptable for publication in *Molecular Systems Biology*.

Dear reviewers,

Thank you very much for your thoughtful comments and giving us the possibility to further improve the article. Below we addressed all comments point by point.

Reviewer #1:

- On page 7, the authors state that some of the gene expression data demonstrate an oscillation pattern, which the authors suggest indicates the presence of indirect effects. However, using cycloheximide to inhibit protein synthesis would be a better way of assessing indirect interactions.**

We agree with the referee that cycloheximide treatment would be an excellent way to evaluate which differentially expressed genes are direct or indirect targets. However, cycloheximide treatment causes to some degree stress to the plants and triggers the upregulation of 18 of the 20 stress-responsive transcription factors (Appendix Figure S23). Because no distinction could be made between the differentially expressed genes as a consequence of the cycloheximide treatment or the induced overexpression of one of the transcription factors, we could not apply such treatment.

We included the cycloheximide data in the supplemental data of this manuscript (Appendix Figure S23) and added the information to the text on page 7, third paragraph: *“To get an indication of which genes are direct or indirect targets of the induced TF, we opted for a time-course approach rather than an inhibition of translation by cycloheximide, because the latter already induced 18 of the 20 TFs by itself (Appendix Fig S23)”*.

Appendix Figure S23 The differential expression of 20 transcription factors upon cycloheximide treatment. With the “differential expression” tool in Genevestigator®, the $\log_2(\text{fold change [FC]})$, $\text{FDR} < 0.05$ was calculated of the 20 transcription factors upon cycloheximide treatment in two experiments (AT-00143 and AT-00208) (Hruz et al., 2008).

- In Figure 3 (described on page 8), the authors group their TF induction data into four waves, categorized based on whether the expression of a gene exceeds a \log_2 (fold change) threshold of greater than 1 at 40 minutes, 1 hour, 2 hours, or 4 hours. However, in figure 3, it looks like many of the genes in wave 1 and wave 2 have very similar profiles, and that most of the genes in these waves reach their maximum at the same time (around 2 hours). This is also true of some of the genes in wave 3. Categorizing these expression patterns by initial rate or expression maximum would be a more useful way to understand the kinetics of the network.**

We agree with the reviewer that this data could be presented in multiple ways and that our choice of presenting or wording might be prone to confusion. With this figure (Fig2 in the revised manuscript), we intend to show that not all transcription factors were induced at the same time, resulting in a sequential manner of upregulation. As pointed out by the reviewer, the categorization of the expression patterns according to initial expression rate would also be a possibility. However, in the setup of this experiment plants were transferred to control or stress conditions and the expression upon stress was compared to the expression levels at control conditions at the same time point. By doing so, potential changes in expression due to the transfer were eliminated. This experimental setup implies that we do not have expression data at time point 0 and cannot calculate the initial expression rate. For these reasons, a categorization based on a threshold appeared, in our opinion, most suitable. However, we agree that term “waves” might not be well-chosen and changed

it to “groups”. We added an additional sentence at page 6, third paragraph, to make the categorization clearer:

“Within 1 h upon stress, 9 of the 20 TF-encoding genes were significantly upregulated (Table EV1; Student’s t-test, FDR<0.05) and most genes reached a maximum expression level after 2 h, demonstrating the very rapid response of this regulatory network. When considering the earliest time points in more detail, the initial upregulation was not equally fast but instead occurred in a sequential manner (Fig 2).”

3. The identities of the 5 core TFs.

We rewrote the abstract which now clearly mentions the 5 core transcription factors.

4. The number of total direct interactions between the 20 TFs identified (I believe this number is 68).

The total number of interactions between a single TF and a target gene is 45 and additionally, 23 direct interactions were uncovered by testing the effect of two TFs simultaneously on a target gene. Thus, the total number of confirmed interactions is indeed 68. We added this number in the results section at page 13, second paragraph and in the discussion at page 18, fourth paragraph.

5. The number of total direct interactions linked to the 5 core TFs (I could not locate this number).

We mentioned the total number of interactions between a single core TF and a target gene, which amounts to 39, in the result section at page 9, second paragraph, and in the discussion at page 15, second paragraph. When evaluating the regulation of a target gene by two TFs, different combinations could confirm the same regulation. For example, the ERF-1 induced upregulation of WRKY15 was confirmed in TEAs when adding ERF9 or STZ. As a consequence, this additional confirmed regulation could be double-classified, once as an interaction with a core TF (ERF9) and as an interaction without a core TF (STZ). For this reason, the total number of interactions linked to the core TFs and a target gene is a difficult number to establish.

6. The maximum change in growth caused by loss or gain of function lines.

ERF59-GR showed the largest increase in rosette area (26%) and ERF6-GR the largest decrease (92%). We added this useful information at page 10, first paragraph: *“The maximum increase and decrease in rosette area was observed in two GOF lines of the core network, ERF59-GR (+26%) and ERF6-GR (-92%), respectively”*.

Reviewer #2:

This work builds on a series of previous publications where the authors have investigated to response to a relatively mild osmotic stress imposed by exposing seedlings to 25 mM mannitol in agar plates. In this condition the plants have reduced growth but otherwise have essentially normal development. The authors have used to system to identify the regulatory mechanisms by which plants restrict their growth under mild osmotic stress. Understanding such growth regulation mechanisms have a number of implications in plant stress biology. Such growth restriction is an adaptive response for plants in many environments; however, being able to turn off or sidestep the growth regulation mechanisms may be a way to improve productivity of crop plants. As part of their previous studies, this group has conducted extensive transcriptome analysis to find differentially regulated genes in the mild osmotic stress conditions. Here they take a group of transcription factor genes found to be upregulated in response to mild osmotic stress and attempt to construct a gene regulatory network (ie: find which of these transcription factors may activate or repress others in this group). They do this via physiological experiments to examine growth and gene expression of their transcription factors in mutant lines lacking activity of a specific transcription factor or in transgenic lines with ectopic expression (driven by 35S promoter) of an inducible form (fusion to glucocorticoid receptor to control entry into the nucleus) of each transcription factor. They also cross their inducible expression lines together to test for additive or antagonistic interactions between transcription factors in their putative network. In another set of experiments, they use transient expression in protoplasts to assay which of their transcription factors can activity expression of other transcription factors in their group. They compare this data an analysis of cis elements in the gene promoters to make sense of the patterns of activation/repression as well as additive and antagonistic interactions among their set of transcription factors. They find that their set of transcription factors do cross regulate each other and there are both additive and synergistic interactions. Some of these interactions are expected based on previous data or analysis of cis-elements but others are not. Thus, it is possible that the transcription factors they analyzed are part of a gene regulatory network. How things are arranged in this network and whether is is a fairly complete network or whether other transcription factors could also be part of this network (for example, factors whose expression is repressed by 25 mM mannitol) is not as clear.

The strength of this work is its scale and ambition. I am not aware of any work in plant stress biology (or perhaps in all of plant biology) that has attempted a comprehensive test of physiological and regulatory interactions between such a large set of genes. Such a work should be of great interest to those working in plant stress biology, as well as a broader audience interested in transcriptional regulation. I say should be as I think in its present form this paper would disappoint a good portion of its readers. This is because there is no overall conclusion that emerges from this work. Many would want to know how we can use the new knowledge of this gene regulatory network to perturb it toward allowing more growth under mild stress or further restricting growth. There is no conclusion and no discussion of this point. The discussion as it is rambling through several topics while making no conclusion. Also a number of things related the network analysis or prediction (such as the equation on page 14 or the usage of "incoherent feedforward loop" on page 16 and elsewhere) are used without any context or explanation. I think that the authors should think carefully of what they are trying to accomplish with this study and who their main audience is (presumably plant stress biologists) and write a paper that is more clear and useful to this audience and in which the main conclusions are more clear. From a technical standpoint, the study suffers from being broad but shallow (see details below) and this also introduces doubt about what can really be concluded from this work.

Major points:

1. In their analysis plants with ectopic expression of transcription factors (Gain of Function-GOF lines in their terminology) they use of only two independent transgenic lines per construct. This problematic as in some cases the two lines give conflicting results. For example, in figure 4C where ERF1-GR has one line with increased growth and one line with decreased growth, ERF2-GR has one line with increased growth and one line with no change, WRDK28-GR has one line with decreased growth and line with no change, WRDK33-GR has one line with

increased growth and one line with no change and WRKY48 has only one line shown. Similar discrepancies can be found in Figure 6C where ERF2, ERF5, ERF11, ERF59 and ERF98 all show dissimilar results between the two transgenic lines. So then how to interpret this data as the whether the ectopic expression has an effect? It seems that for their analysis of cell number and cell area in figure 4 E and F (and also presumably for the crosses conducted later in the study) they just pick the one transgenic line that has an effect (I do acknowledge them for clearly labeling which transgenic line was analyzed). But then how to know whether this is a real effect of the transgene rather than a mutation(s) introduced during transformation? It is even more confusing as they do check that the two transgenic lines have similar levels of gene expression. Such a shallow analysis and just picking the one transgenic line that shows you what you want to see would not be allowed in a more targeted study of a few genes and is not sufficient here either. They need to have 3 or more independent transgenic lines per construct to know what the real effects are. I suspect this issue also affects their gene expression results but it is more difficult to parse out whether the data shown are from both transgenic lines or just the one that showed a difference.

We acknowledge the comments of the reviewer and therefore phenotypically analyzed, where possible, a third independent GOF line of the transcription factors. For some transcription factors only two independent lines were available and generating new lines would delay the publication of this work by about one year. No new overexpression lines for ERF5-GR, ERF6-GR and ERF11-GR were analyzed as these transcription factors and their phenotypes when overexpressed were subject to previous publications (Dubois et al., 2013; Dubois et al., 2015). In total, we performed leaf series of an extra 14 lines (including control lines) in three or four independent repeats and added this data in the revised manuscript. We included the additional phenotypic data in the main figures of the manuscript (Fig 4 and Fig 6) and in the appendix (Appendix Fig S3, S7, S8, S10, S11, S17, S20) and adjusted the text of the result section at page 9, 10, 11 and 12. We also added the overexpression levels of the new lines in the appendix (Appendix Fig S3, S7, S8, S10, S11, S17, S20).

The additional phenotyping confirmed the growth-repressing function of ERF9, ERF98 and WRKY15. The third independent line of ERF8 did not show a growth phenotype, as was the case for the first and second independent line. However, for ERF-1 and WRKY33 the analysis of a third independent line did not lead to a conclusion regarding their growth-regulating function because all three lines show a conflicting phenotype (Figure 4C-6C). In conclusion, six overexpression lines did not show a growth phenotype (ERF8, RAP2.6L, STZ, ZAT6, MYB51 and WRKY48), five TFs were identified as growth repressors (ERF6, ERF9, ERF11, ERF98 and WRKY15) and one was identified as a growth activator (ERF59). For the other five overexpression lines (ERF-1, ERF2, ERF5, WRKY28 and WRKY33) no conclusive results were obtained regarding their growth-regulating function. For ERF9 and ERF59, two of the three independent lines showed a significant increased sensitivity to mannitol stress. The three independent lines of ERF-1, ERF8, ERF98 and WRKY15 did not show a sensitivity phenotype (Fig 4C and Fig 6C).

Figure 4C-6C – Phenotypic analysis of the gain-of-function (GOF) lines of the TFs.

4C Rosette area of two or three independent GOF lines per TF (calculated as the projected area or the sum of the leaves, $n = 10$ plants) germinated and grown on DEX-containing medium. Measurements were performed at 22 DAS relative to the control line. “Independent line 1” is the line with the highest overexpression level.

6C Rosette area of two or three independent GOF lines ($n = 10$ plants) germinated on DEX-containing control or mild osmotic stress medium was measured at 22 DAS and the reduction on mild osmotic stress was calculated relative to the control line. “Independent line 1” is the line with the highest overexpression level.

Data information: The green bars represent the additional phenotyped GOF lines. Data are presented as mean \pm SEM, $n = 3$ independent experiments, $*=P < 0.05$ (Tukey’s test).

We would like to emphasize that we only drew conclusions about the genes that had two/three independent lines with a similar phenotype. Indeed, for the cellular analysis, we analyzed the line with the clearest leaf phenotype, because we had no motive to also draw the cells of lines that did not show a growth phenotype at leaf level. While this might be useful for one-gene studies to look for potential compensation mechanisms, it was rather impossible to do so in our large-scale analysis. For the crosses between the five GOF lines of the core network, it should be noticed that both lines showed the same growth phenotype. We thus chose the line that was used for the expression analysis. Finally, regarding the concern about the similar expression level in the lines with different phenotypes, we point out that we analyzed a line with a high overexpression level and another with

a moderate overexpression level, as stated on page 7, third paragraph. In some cases, the $\log_2(\text{FC})$ values do not differ greatly between the two lines but small differences between $\log_2(\text{FC})$ values do result in a large difference between the actual values. We regret that the manuscript might have left a somewhat a shallow impression and therefore added a table (Appendix Table S2) indicating the lines chosen for expression analysis and crosses.

Appendix Table S2 – Overview of the GOF lines used for the large-scale expression analysis and the crosses.

A large-scale expression analysis was performed in which putative direct and indirect targets were identified. The independent GOF lines used for this analysis and used for the crosses is presented in the table. NA = Not Applicable.

Gene	GOF line used for expression analysis	GOF line used for cross?
ERF-1	#2	
ERF2	#1	
ERF5	#1	
ERF6	#1	#1
ERF8	#2	#2
ERF9	#2	#2
ERF11	#2	
ERF59	#1	#1
ERF98	#1	#1
RAP2.6L	#1	
STZ	#1	
ZAT6	#1	
MYB51	#1	
WRKY6	NA	
WRKY15	#2	
WRKY28	#2	
WRKY30	NA	
WRKY33	#2	
WRKY40	NA	
WRKY48	#1	

2. **Second, they use protoplast transient expression assays (TEA assays in their terminology) to test cross regulation among different transcription factors in their putative network. While this can be a useful assay, it does have limitation and their interpretation of such data stretches beyond the limitations of this assay. These assays force high level expression of the transcription factor and thus can lead to reporter activation by transcription factors that would not normally activate the reporter in planta. Indeed, they themselves say (bottom of page 7) that the results of the stable transgenic lines expressing transcript factor-GR receptor fusions and transient expression assays do not always agree. They make the conclusion that the transient expression assays more likely represent the actual endogenous activity of the transcription factor. This is not true, or not necessarily true, because of the enforced high level expression in the protoplast assays and because protoplasts are already stressed cells prone to different responses than an intact plant.**

We agree with the reviewer that protoplast transient expression assays have their limitations and might cause false positives due to enforced expression levels. We were aware of this concern, and therefore paid particular attention to it: all TEAs were performed in 3 independent experiments, each including 4 biological repeats. Moreover, we emphasize that we only validated the regulatory link when (1) the TEA gave a positive result, and (2) the target gene was significantly upregulated within 4h induced overexpression of the upstream TF. The large majority of the transcription factors did not show discrepancy between the protoplast assay and the response in the GOF lines.

The nature of regulation was different when it concerned transcriptional repressors, such as ERF8 and ERF9. Both genes contain the well-described ERF amphiphilic repressor (EAR) domain. Literature describes these genes as repressors and our TEAs also confirm their repressing activities. To our surprise, ERF8-GR and ERF9-GR caused upregulation of the target genes *in planta*. We investigated this discrepancy and performed additional TEAs which showed that the C-terminal GR-tag influences the transactivation capacities of the transcriptional repressor (Appendix Figure S24). The discrepancy is thus most likely due to the presence of the GR-domain close to the EAR-motif, rather than being an intrinsic problem of the TEA assays. In these cases, the TEAs performed with the TFs without GR domain, are thus more likely to represent the endogenous situation *in planta*. We have modified the text at page 8 and 9, last and first paragraph respectively, to better describe these observations and interpretations:

“The repressing function of the in literature-described repressors ERF8 and ERF9 (Ohta et al, 2001; Nakano et al, 2006) seemed to be abolished by fusion with the GR-domain, as observed in planta (Appendix Fig S3-S22) and in TEAs performed with ERF8-GR or ERF9-GR (Appendix Fig S24). The discrepancy is thus most likely due to the presence of the GR-domain close to the EAR-motif. The TEAs performed with the TFs without GR domain are thus more likely to represent the activity of the endogenous TF.”

Appendix Figure S24 – Transient expression assays to assess the addition of a GR-domain.

A, B, C The effect of ERF8 (dark green) and ERF8ΔEAR (light green) (A), of ERF9 (dark green) and ERF9-GR (light green) (B) and of ERF59 (dark green) and ERF59-GR (light green) (C) on target genes identified in this study, after addition of 10 μM DEX for 4 h. The relative luminescence was calculated relative to the control, 35S::GUS (gray). n = 4 biological repeats. Data information: Data are represented as mean ± SEM, n = 3 independent experiments

- Also, they also do transient expression experiments where two of the transcription factors are expressed at once. This leads to various kinds of effects which they try to interpret based on looking at the pattern of cis-elements in the promoters of their genes. They even say (last paragraph on page 13) that they use such search approach to "test whether two TF compete for binding sites in the promoter of a shared target gene". This is flawed. Even though these transcription factors come from relatively well characterized families and their core recognition sites are known, their actual promoter binding sites (which can be influenced by sequences flanking the core recognition sites, epigenetic regulation, other promoter binding proteins and other factors) are not known. And, some of the interaction effects they observed could be explained by different protein levels of the two transcription factors (protein levels are not measured). If they want to talk about promoter binding or competition for promoter binding, then they need to actually measure promoter occupancy of their transcriptions factors using intact plants.

The whole paragraph at the bottom of page 13 is at best a side point for the discussion as it is speculation and not based on actual data.

Thank you for pointing this out. We agree that this analysis is not sufficient to determine competition between two TFs. We wanted to provide a possible mechanism for the different situations observed in the TEAs performed with two TFs. The assessment of the motifs in the 20 promoters is a bioinformatical analysis of precise motifs extracted from experimental data and is thus not speculation. We moved this section to the discussion (at page 15 and 16) and tuned down the mechanistical interpretation of the observed effect.

- 4. Related to this concern, it must be pointed out that their conclusions about direct versus indirect regulation are based mainly on time course data of which factor is induced first, which one a bit later etc. The networks they draw and the edges they populate them with are based only on expression changes. There is no data in this study of who is actually binding to whose promoter. They do acknowledge (bottom of page 6 and page 7) that such connections could be direct or indirect and that their putative network contains many indirect connections. But then they seemingly forget about this distinction in other parts of the manuscript. The transient expression assays are conducted in plant cells so indirect effects are possible-in other words their introduced transcription factors could activate the reporter construct via mechanisms other than directly binding to its promoter. Here again they need to either measure promoter occupancy or more clearly acknowledge the limitations of the assays they use and be appropriated cautious in their conclusions.**

We are aware of the concern of the referee and tuned down the conclusions in the results (page 8, 11 and 13) as well in the discussion (page 16, 17 and 18). To further strengthen the robust network, we searched for datasets that did perform promoter occupancy experiments. A previously performed DNA affinity purification sequencing (Dap-seq) resulted in the generation of a large-scale dataset of genome-wide binding locations of 529 TFs (Barlett et al., 2017). This technique is based on affinity-tagged TFs expressed *in vitro* and the construction of gDNA-derived libraries. Due to technical issues and low protein expression only ~30% of the 1,812 Arabidopsis TFs passed quality thresholds. Unfortunately, there was no data available for ERF6, ERF9, ERF59, ERF98, ZAT6 and WRKY48. For the other TFs, this dataset validated the promoter binding of 9 out of 16 confirmed TEA regulations. We are aware that additional *in planta* validation of the not yet confirmed regulatory links would be suitable, but unfortunately, approaches as ChIP or EMSA are not possible given the large number of TFs and the limited timing given for preparing this rebuttal. We discuss this as possible perspectives in the discussion. We added this information in an extra column of the Appendix Table S3.

Appendix Table S3 – Confirmed regulatory interactions between 20 genes encoding transcription factors.

The putative direct regulatory interaction that were confirmed with transient luciferase assays (TEAs). A large-scale expression analysis was performed in which putative direct targets were identified. The measured expression values, the FDR corrected p-values, the first time point at which the gene is significantly differential expressed from this expression analysis, the average luminescence and the results from a large-scale promoter occupancy dataset (O'Malley et al., 2016) are given. FC = Fold change.

Promotor	Regulator	Log ₂ (FC)	FDR	Time point	TEA	DAP-seq
ERF1	ERF59	0.90	0.019103	1h	1.49	NA
ERF1	ERF8	0.61	0.016902	1h	0.79	✓
ERF1	ERF98	0.62	0.071438	1h	2.37	NA
ERF1	WRKY48	0.94	0.020846	1h	1.36	NA
ERF11	ERF59	2.68	0.070405	1h	15.38	NA
ERF11	ERF8	2.12	0.028033	1h	0.58	✓
ERF11	ERF9	2.40	0.041776	1h	0.90	NA
ERF5	ERF8	0.79	0.020526	1h	0.55	✓
ERF5	ERF9	0.86	0.044109	1h	0.85	NA
ERF5	ERF98	1.36	0.040165	1h	1.43	NA
ERF59	ERF6	2.03	0.01408	1h	1.47	NA
ERF59	ERF98	2.24	0.050572	1h	1.35	NA
ERF6	ERF59	2.43	0.019147	1h	1.52	NA
ERF6	ERF8	1.90	0.030811	1h	0.70	✓
ERF8	ERF9	-0.57	0.077533	2h	0.46	NA
MYB51	ERF9	0.79	0.098944	1h	0.56	NA
MYB51	ERF6	1.20	0.001165	4h	1.36	NA
RAP26L	ERF98	2.98	0.003078	1h	1.42	NA
RAP26L	WRKY15	2.60	0.094323	2h	1.22	✓
RAP26L	ERF8	2.91	0.030273	2h	0.74	✓
STZ	ERF59	3.22	0.006443	1h	1.68	NA
STZ	ERF8	3.31	0.000317	1h	0.48	×
STZ	ERF9	2.77	0.005192	1h	0.43	NA
STZ	ERF6	1.91	0.09159	4h	1.15	NA
STZ	ZAT6	1.87	0.065485	1h	0.49	NA
STZ	ERF2	3.53	0.025831	4h	1.38	×
STZ	ERF11	2.12	0.015696	2h	0.82	×
STZ	WRKY15	2.75	0.008891	2h	1.32	✓
WRKY15	ERF8	0.97	0.003826	1h	0.73	✓
WRKY30	ERF98	3.23	0.078848	1h	1.52	NA
WRKY30	ERF59	-3.62	0.079938	4h	1.65	NA
WRKY33	ERF8	1.03	0.040496	1h	0.71	×
WRKY33	ERF59	1.27	0.045276	1h	1.87	NA
WRKY40	ERF8	1.63	0.027366	1h	0.64	×
WRKY40	ERF9	1.61	0.045276	1h	0.77	NA
WRKY40	ERF98	2.28	0.016902	1h	1.24	NA
WRKY48	ERF9	2.33	0.011796	1h	0.59	NA
WRKY48	ERF98	3.38	0.001349	1h	1.39	NA
WRKY48	ERF8	2.54	0.006443	1h	0.40	×
WRKY6	ERF8	0.87	0.01733	1h	0.60	✓
WRKY6	ERF9	0.72	0.046289	1h	0.78	NA
WRKY6	ERF98	1.21	0.0013	1h	1.20	NA
ZAT6	ERF8	1.80	0.033792	1h	0.50	×
ZAT6	ERF9	1.56	0.08971	1h	0.77	NA
ZAT6	ERF59	2.19	0.079281	2h	2.20	NA

- 5. It seems that their transgenic plants expressing transcription factor GR-receptor domain fusions were germinated and growth on continuous supply of DEX and the final plant size measured. How to rule out effects of some lines germinating more quickly, or slowly, than others? This also seems to nullify some of the advantages of making transgenic lines with inducible activation of the transcription factor.**

To assure equal germination, we include a stratification period of 48h and verified visually that the lines germinated on the same day. Additional, we took pictures of every plate from 9DAS until 22DAS. The pictures at early time points enabled us to rule out that large differences in germination were present amongst the analyzed lines.

For the phenotypic analysis, we opted for the germination on DEX because, according to us, this was the most physiologically relevant way to assess the growth of these lines: there was no need for transfer, no mesh or liquid culture which could adversely affect growth.

- 6. It also must be pointed out that no outgroup of transcription factors analyzed. Most of the effects on growth they observed are relatively subtle (the previously studied ERFs which are also included here are an exception). How do we know that similar effects would not be observed just by randomly picking and analyzing members of the ERF or WRKY gene families (or other types of transcription factors)?**

Indeed, no outgroup of transgenic lines of other TFs was analyzed. Our aim was not to show that the TFs we selected are the only ones involved in growth regulation. Growth is a very complex trait and is probably regulated by many different pathway and there is no doubt that additional TFs, not part of this network, are also involved in leaf growth regulation. We demonstrated here that growth is not regulated by a simple linear pathway but by a network with many regulatory connections. Thus, redundancy within this network results in mild or no phenotypes when one gene is knocked-out. However, the overexpression of a gene does render strong phenotypes, previously not published. For example, ERF59-GR had an increase in rosette area of 26% and ERF9-GR a reduction of 58%.

Specific comments (some of these are more minor issues, some are specific examples of the concerns expressed above)

- 1. Page 8: The "GRN is activated in four waves" section seems out of place here and would seem better presented earlier in the manuscript as preliminary data which helps to interpret the other analysis of transgenic plants.**

This is an excellent suggestion. We moved this section after “A GRN of 20 TFs is specifically induced in growing leaves exposed to mannitol” and before the “The GRN is highly interconnected and dynamic”.

- 2. Figure 4A: it seems that their analysis of T-DNA mutants (LOF lines in their terminology) includes only one T-DNA allele for each gene. As explained above for the limited number of independent transgenic lines assayed, this would be considered problematic in more targeted studies and must be kept in mind when interpreting this data also. There is no mention in the manuscript of whether the T-DNA mutants and transgenic GOF lines have opposing effects on growth or gene expression as would be expected. For example, the wrky15 T-DNA line has reduced growth (Fig 4A) and the WRKY15-GR overexpression line also has reduced growth (Fig 4C)-how to explain this?**

This is indeed a topic that should be addressed. We now included a table (Table EV2), listing the different growth phenotypes in control conditions. The table gives a simplified overview for the reader to quickly assess whether e.g. LOF and GOF line give opposite phenotypes. In addition, the table reports whether we addressed the TF as a growth promoting or repressing protein.

Regarding the apparent discrepancy about WRKY15, it should be noticed that in this case both GOF lines and the third GOF line gave the same phenotype (reduced growth), so the described phenotype is robust. Also, we performed cellular analysis in both the LOF line and the GOF line, and in both cases, we observed that the growth reduction was due to a decrease in cell number, although more pronounced in the LOF line. Thus, these effects are surprising but real and suggest that this transcription factor needs to be present with an optimal activity as also observed for concentration dependent effects of plant hormones. We added a remark in the discussion at page 20, second paragraph:

“For WRKY15 and ERF59, the GOF lines in this study and their previously studied overexpression lines (under the control of the 35S promoter) (Pré et al, 2008) showed a contrasting phenotype. In both cases, it is likely that the overexpression level is crucial and we could hypothesize that the growth-promoting or growth-repressing function of the TF depends on an expression optimum.”

Table EV2 - Overview of the phenotypic analysis of the 20 TFs under normal conditions.

The outcome of the phenotypic analysis of the two or three independent GOF lines and the LOF line is given. A plus (+) and minus (-) represents a significant larger and smaller rosette area at 22DAS compared to the control, respectively. If no significant changes of the rosette area were observed a "0" is indicated and when a line was not present "NA" (Not Applicable) was indicated. The final outcome of the growth function of the TF is given considering that at least two independent GOF lines or the LOF line showed a growth phenotype and that the GOF and LOF lines did not show a contrasting growth-regulating function.

Gene	Phenotype GOF line			Phenotype LOF line	Growth function
	#1	#2	#3		
ERF-1	-	+	0	0	Inconclusive
ERF2	+	0	NA	-	Activator
ERF5	0	-	NA	0	Inconclusive
ERF6	-	-	NA	0	Repressor
ERF8	0	0	0	+	Repressor
ERF9	-	-	-	0	Repressor
ERF11	-	-	NA	+	Repressor
ERF59	+	+	0	0	Activator
ERF98	-	-	-	0	Repressor
RAP2.6L	0	0	NA	-	Activator
STZ	0	0	NA	-	Activator
ZAT6	0	0	NA	0	None
MYB51	0	0	NA	0	None
WRKY6	NA	NA	NA	-	Activator
WRKY15	-	-	-	-	Inconclusive
WRKY28	-	0	NA	0	Inconclusive
WRKY30	NA	NA	NA	+	Inconclusive
WRKY33	-	+	0	0	Inconclusive
WRKY40	NA	NA	NA	0	Inconclusive
WRKY48	0	NA	NA	0	Inconclusive

3. Page 11, authors state "To evaluate whether the observed changes in GA2-OX6 expression were a direct or indirect effect of the overexpressed TF, the binding of the TF on the 2-kbp GA2-OX6 promoter was evaluated with TEAs in tobacco protoplasts." This is incorrect, the transient expression experiments do not assay promoter binding (there is no data on promoter binding in this manuscript).

We are grateful that you pointed specific examples of our incorrect use of the terms "promoter binding", "direct targets", etc. We rephrased this section and used for example "transactivation capacities" and "confirmed targets".

4. Top of page 12: why is this normalization scheme for the growth data used here and not in other experiments such as those presented in Figure 4?

We opted for this normalization scheme because we wanted to show the mannitol-induced reduction in rosette area and not the absolute rosette area. The mannitol-induced reduction gives a clearer representation of the responsiveness of a transgenic line to stress. The absolute rosette areas under might be misleading because e.g. a line with a smaller rosette compared to the wild type under mannitol conditions will appear to be more sensitive, however, when this line is already smaller under normal conditions, the relative reduction by mannitol is the same as in the wild type and thus shows no altered sensitivity. The normalization scheme represents in our opinion better this nuance of sensitivity to stress. In the appendix, the absolute leaf areas can be found (Appendix Figure S3-S22).

5. Page 14, The authors state: "The expected rosette area of the double cross was calculated based on the rosette area of both parental single crosses according to the following formula: $\log_2(\text{expected double cross}) = \log_2(\text{single cross 1}) + \log_2(\text{single cross 2}) - \log_2(\text{control})$ and compared with the observed area via the chi-square test (Vanhaeren et al, 2014)." How is this formula derived and what is its relevance here? And, it seems the result of this comparison between observed and predicted growth is

not actually presented as it is not shown in Figure 8 (note that I cannot find figure EV2 in my review material).

To determine genetic interaction, two different models were previously applied: an additive model or a multiplicative model (Fisher et al, 1918 and Phillips et al., 2009). In an additive model, the effects of the single crosses are summed and compared to the reference (here, GFP-GR). With a multiplicative model, the effects of the single crosses are multiplied and compared to the reference. In this manuscript and previously in Vanhaeren et al., 2014 (eLife) we opted for the multiplicative model because it is more stringent. The formula allows us to explore the genetic interaction between the two genes that were combined in the cross. We can evaluate whether the observed phenotype of the double cross was expected which could indicate a combination of genes that do not interact genetically. When the observed phenotype was unexpected, the genes involved in the combination are potentially part of the same pathway and thus interact genetically. In this case, we could evaluate whether the cross resulted in positive or negative interaction. Positive interaction, in this manuscript results in a synergistic phenotype and means that the phenotype is larger than expected. Negative interaction means that the phenotype is smaller than expected and resulted in a negative phenotype.

As advised by our bio-statistician, we performed a new statistical analysis using a statistic model that takes into account the variability on the average rosette areas. We added a full explanation in the materials and methods section at page 23. This slightly modified the outcome of the statistical tests, and we adapted and discussed this appropriately in the revised version at page 14 in the results and at page 19 in the discussion. To avoid confusion, we moved the formula from the main text to materials and methods. We also added the expected rosette area to the graphs in Figure EV2 to improve clarity.

Figure EV2 – Phenotypic analysis of crosses between two gain-of-function (GOF) lines of the core network.

A, B, C All members of the core network were crossed with either each other, referred to as double cross, or with the control line (GFP-GR), referred to as single cross and were germinated and grown on dexamethasone (n=8 plants). At 22 DAS the projected rosette area of the double and single cross and the GFP-GR was measured. Double crosses resulted in a synergistic (A), an additive (B) or a negative (C) phenotype and are depicted in orange, green and blue connection, respectively. Data information: Data are presented as mean \pm SEM, n = 3 independent experiments.

- 6. Page 16: What is an "incoherent feedforward loop"? The following sentence "When looking at the network globally, an incoherent feedforward loop seems to be at the base of the adaptation mechanism (see below)" adds nothing. As stated above, consider what your audience is and what the main conclusion is and try to present things more clearly (maybe I am not the intended audience).**

Thank you for pointing out this ambiguity. Since our intended audience reaches beyond systems biologists, we will pay particular attention in making this comprehensive. An incoherent feedforward loop is a network motif in which, starting from the input node, two pathways converge to the same output node. One pathway leads to an inhibitory effect on the output node, the second pathway to an activation effect. Because this manuscript is studying a network and we want to link the network to its biological function, how nodes and edges are connected to each other (= network topology) is important. Previously performed research has shown that only two types of network motifs can lead to adaptation (as explained on page 16, last paragraph). Adaptation is defined as "*the ability of circuits to respond to input change but to return to the prestimulus output level, even when the input change persists*" (cited Ma et al., 2009). This network contains one of these two network motifs, namely the incoherent feedforward, which we later on discuss in more detail (at page 16 and 17). However, we agree that this terminology needs more explanation and added additional information on incoherent feedforward loops at page 16.

- 7. Likewise, having section headings of "topology of the network" and "wiring of the network" is just obfuscates things.**

We rewrote the discussion thoroughly and changed the headings to, for us, more clear and straightforward titles.

- 8. Page 16-17, this paragraph about post translational modification is OK as you clearly state that it is speculation; however, the fact that you actually drew phosphorylation into your model in Fig 9 is not as if presenting this as fact when you don't actually have such data.**

We agree and removed the phosphorylation nodes in figure 9.

- 9. Page 17 authors state " Because the network genes encode relatively small proteins, we speculate that the time between the detection of the mRNA transcript with e.g. qPCR, the translation and folding of the protein and the subsequent induction of the target gene is less than half an hour" This statement is just silly, translational rate and preferentially translational happen and matter a lot more than just the size of the protein.**

Indeed, a lot more factors contribute to the translational rate than just protein size. This sentence was included to communicate that it is possible for genes of the first wave to induce genes of the second wave. To avoid confusion, we removed the sentence.

- 10. Page 17: "Our TEAs have shown that two TFs could form combinations to regulate their downstream targets." You do not show anything about transcription factors forming combinations. Please be more careful with your wording.**

We agree that this wording might be prone to confusion. We rephrased this sentence into "*We also discovered that the combined regulation of two TFs on the same target gene in our TEAs does not necessarily lead to an additive effect of the single regulations, increasing the complexity even more.*" on page 15, second paragraph.

- 11. Page 18: "45 direct regulatory interactions were experimentally validated with TEAs and when combining two TFs," Please note the above stated concerns about calling**

these direct interactions in the absence of promoter occupancy data and based solely protoplast transient expression.

We fully agree and rephrased it now as “*In addition to the 45 confirmed regulatory interactions, another 23 regulations could be corroborated when co-transforming two TFs, adding to in total 68 confirmed interactions and leading to the observation that for the induction of some downstream TFs, a set of TFs is necessary.*” on page 17, last paragraph.

12. Page 18: "and is hereafter referred to as the core network." You have referred to the core network many times before this statement. Are you now referring to a different core network?

We were not referring to a different core network and apologize for this mistake from our part. We corrected it in the manuscript.

13. Bottom of page 18: here again "incoherent feedforward loop" with no explanation of what this is or how it is relevant.

We agree and refer to our reply on comment #6. We explain the meaning and relevance of this incoherent feedforward loop at page 16.

14. Page 20-21: this part is even more rambling than the rest of the discussion. What is your main point?

Our main point is that the transcription factors studied in this manuscript might be part of a more general stress response and we provide literature data to support this. The upregulation of the TFs has been detected upon multiple stresses. However, whether the upregulation is specific to growing leaf tissue requires more research.

We have carefully rewritten the discussion to make it more comprehensive and better structured.

15. Bottom of page 21: "indirect strong repression of WRKY30". Here again, you cannot say for sure what is direct and indirect.

We agree and removed this sentence.

16. Overall, I find the discussion very rambling and not clearly state what the authors view as the main and most novel findings of this study. The discussion should be shortened and restructured.

We rewrote the discussion taking your comments into account. We now clearly stated our main findings and their importance for further research:

- **Linear pathways are a simplification.** We have shown here that multiple TFs can regulate the same target genes and that in some cases more than one TF is necessary to induce the expression of a target gene.
- The network is **robust** because regulatory redundancy is built in, making the network less susceptible to mutations. ERF6 and ERF98 are both induced in the first induction group and can transcriptionally activate a large part of the network. They have an overlap of 6 target genes, considering all TEAs. ERF8 and ERF9 are both induced in the third induction group and can transcriptionally repress a large part of the network, showing an overlap of 9 target genes.
- The network is efficient for **environmental adaption** to a stress signal. The repressing activities in the network after 2 hours of stress enables the network to return to its prestimulus state.
- The network is highly **responsive** to a range of input signals and might be part of a general stress response. However, the need for two TFs to activate targets **prevents stochastic activation** of the network. The random induction of the network would lead to a needless stress response which is disadvantageous for the plant.

17. There are cases throughout the paper where there are small grammar errors (tenses and subject verb disagreement for example or missing verb, incorrect phrases) and editing of the paper would improve clarity.

Thank you for pointing this out to us. We evaluated the text for grammar errors very carefully.

2nd Editorial Decision

18 November 2017

Thank you again for submitting your work to Molecular Systems Biology. We are now satisfied with the modifications made and will be able to accept your paper for publication in Molecular Systems Biology pending the following minor amendments:

*Please add a conflict of interest statement at the end of the manuscript

*We would be grateful if you could reformat the Data Availability Section as follows:

"

#Data and software availability

- nCounter nanostring experiments in wild-type plants: ArrayExpress, E-MTAB-6205

(<https://www.ebi.ac.uk/arrayexpress/experiments/E-MTAB-6205/>)

- nCounter nanostring experiments in 17 GOF lines: ArrayExpress, E-MTAB-6209

(<https://www.ebi.ac.uk/arrayexpress/experiments/E-MTAB-6209/>)

"

Please note that we could not find the datasets under the above numbers. We would be grateful if you could confirm the links and the release of the data as soon as the paper will be accepted.

*Since Table EV1-3 are relatively small, it is OK to include them as PDF. Please move their respective legends from the manuscript file to the corresponding EV table files so that the legends are directly visible.

*Callouts

- Please add an explicit callout to fig. 9C in the main text.

*Source data

- If possible, please split the file called 'Figure 1-2 Source Data' in one file per figure. Otherwise, if you prefer to keep this dataset as a single file, it should then be renamed Dataset EV1 and referred to from the main text as "Dataset EV1".

*Figures

- The labels on Figure 4B and 6B are a little hard to read. The figure would be more legible if the font would be larger.

2nd Revision - authors' response

25 November 2017

The authors made the requested editorial changes and submitted the final version of their manuscript.

Corresponding Author Name: Prof Dirk Inze

Manuscript Number: MSB-17-7840RR